# Tools for Verifying Neural Models' Training Data

**Dami Choi**[*]

U. Toronto & Vector Institute

choidami@cs.toronto.edu

**Yonadav Shavit**[*]

Harvard University

yonadav@g.harvard.edu

**David Duvenaud**

U. Toronto & Vector Institute

duvenaud@cs.toronto.edu

## Abstract

It is important that consumers and regulators can verify the provenance of large neural models to evaluate their capabilities and risks. We introduce the concept of a "Proof-of-Training-Data": any protocol that allows a model trainer to convince a Verifier of the training data that produced a set of model weights. Such protocols could verify the amount and kind of data and compute used to train the model, including whether it was trained on specific harmful or beneficial data sources. We explore efficient verification strategies for Proof-of-Training-Data that are compatible with most current large-model training procedures. These include a method for the model-trainer to verifiably pre-commit to a random seed used in training, and a method that exploits models' tendency to temporarily overfit to training data in order to detect whether a given data-point was included in training. We show experimentally that our verification procedures can catch a wide variety of attacks, including all known attacks from the Proof-of-Learning literature.

## 1 Introduction

How can we verify the capabilities of large machine learning models? Today, such claims are based on trust and reputation: customers and regulators believe that well-known companies building AI models wouldn't lie about the training data used in their models. However, as the ability to build new AI models proliferates, users need to trust an ever-larger array of model providers at their word, and regulators may increasingly face malicious AI developers who may lie to appear compliant with standards and regulations. Worse, countries developing militarily-significant AI systems may not trust each others' claims about these systems' capabilities, making it hard to coordinate on limits.

AI developers can enable greater trust by having a third party verify the developer's claims about their system, much as the iOS App Store checks apps for malicious code. Current black-box approaches to model auditing allow some probing of capabilities [Cen23], but these audits' utility is limited and a model's capabilities can be hidden [GTB22, GKVZ22]. An auditor can more effectively target their examination if they also know the model's training data, including the total quantity, inclusion of data likely to enable specific harmful capabilities (such as texts on cyber-exploit generation), and inclusion of safety-enhancing data (such as instruction-tuning [OWJ+22]). However, if auditors rely on AI developers to self-report the data, developers could falsify their reports. This uncertainty limits the trust such audits can create.

In this work, we define the problem of Proof-of-Training-Data (PoTD): a protocol by which a third-party auditor (the "Verifier") can verify which data was used by the developer (the "Prover") to train a model. Our verification procedures assume that the Verifier can be given access to sensitive information and IP (e.g., training data, model weights) and is trusted to keep it secure; we leave the additional challenge of simultaneously preserving the confidentiality of the training data and model weights to future work. In principle, one could solve PoTD by cryptographically attesting to the

---

[*]Equal contribution

37th Conference on Neural Information Processing Systems (NeurIPS 2023).

results of training on a dataset using delegated computation [CKV10]. However, in practice such delegation methods are impractically slow, forcing us to turn to heuristic verification approaches.

Inspired by the related literature on "Proof-of-Learning" (PoL)[JYCC+21], we propose that model-trainers disclose a *training transcript* to the Verifier, including training data, training code, and intermediate checkpoints. In Section 4, we provide several verification strategies for a Verifier to confirm a training transcript's authenticity, including new methods that address all published attacks in the Proof-of-Learning literature. We demonstrate the practical effectiveness of our defenses via experiments on two language models (Section 6). Our methods can be run cheaply, adding as little as 1.3% of the original training run's compute. Further, we require no change to the training pipeline other than fixing the data ordering and initialization seeds, and storing the training process seeds for reproducibility. Still, like PoL, they sometimes require re-running a small fraction of training steps to produce strong guarantees.

The verification strategies we describe are not provably robust, but are intended as an opening proposal which we hope motivates further work in the ML security community to investigate new attacks and defenses that eventually build public confidence in the training data used to build advanced machine learning models.

## 2 Related Work

We build on [Sha23], which sketches a larger framework for verifying rules on large-scale ML training. It defines, but does not solve, the "Proof-of-Training-Transcript" problem, a similar problem to Proof-of-Training-Data that additionally requires verifying hyperparameters.

**Proof-of-Learning.** [JYCC+21] introduce the problem of Proof-of-Learning (PoL), in which a Verifier checks a Prover's ownership/copyright claim over a set of model weights by requiring the Prover to prove that they did *at least as much computational work* as was required to have trained the original model. The Prover's proof consists of reporting a sequence of valid weight checkpoints that led to the final weights. The Verifier then re-executes training between a few checkpoints and checks that the results are similar to confirm (or reject) the report's correctness. Unlike PoL, Proof-of-Training-Data requires proving *what exact data* was used to produce a model. This is strictly harder: any valid PoTD protocol can serve as a solution to PoL, as compute cost can be derived from the size of the dataset. Furthermore, PoL only requires robustness to adversaries that use less computation than the original training run, whereas PoTD targets all computationally-feasible adversaries.

Several works have successfuly attacked the original PoL scheme; our proposed defenses defeat all published attacks. [ZLD+22] and [KRCC22] provide methods for forging fake data that will (during retraining) interpolate between chosen checkpoints, thus allowing the production of fake transcripts. (See Section 4.3 for our mitigation.) [FJT+22] exploit the fact that, due to the prohibitive cost of retraining, the Verifier can only retrain a tiny fraction of checkpoints. By falsifying only a modest fraction of checkpoints, a Prover can prevent their fake transcript being caught with high probability. (See Section 4.2 for our mitigation, which introduces a method for verifying checkpoints that is much cheaper than retraining, and can thus be applied to *all* checkpoints.)

**Memorization during training.** [ZIL+21] introduce the notion of counterfactual memorization (the average difference in model performance with and without including a specific point in training) that is most similar to our own, and use it to investigate different training points' effects on final model performance. [FZ20] examine which datapoints are most strongly memorized during training by using influence functions, but they focus on the degree of memorization only at the end of training. [BPS+23] show that per-datapoint memorization of text (as measured by top-1 recall) can be somewhat reliably predicted based on the degree of memorization earlier in training. [KGG+22] analyze pointwise loss trajectories throughout training, but do not focus specifically on the phenomenon of overfitting to points in the training set.

## 3 Formal Problem Definition

In the Proof-of-Training-Data problem, a Prover trains an ML model and wants to prove to a Verifier that the resulting target model weights $W^*$ are the result of training on data $D^*$. If a malicious Prover used training data that is against the Verifier's rules (e.g., terms of service, regulatory rules) then that

Prover would prefer to hide $D^*$ from the Verifier. To appear compliant, the Prover will instead lie and claim to the Verifier that they have used some alternative dataset $D \neq D^*$. However, the Prover will only risk this lie if they believe that with high probability they will not get caught (making them a "covert adversary" [AL07]). The goal of a Proof-of-Training-Data protocol is to provide a series of Verifier tests that the Prover would pass with high probability if and only if they truthfully reported the true dataset that was used to yield the model $W^*$.

Let $D \in \mathbb{X}^n$ be an ordered training dataset. Let $M$ contain all the hyperparameters needed to reproduce the training process, including the choice of model, optimizer, loss function, random seeds, and possibly details of the software/hardware configuration to maximize reproducibility.

**Definition 1.** *A valid Proof-of-Training-Data protocol consists of a Prover protocol $\mathcal{P}$, Verifier protocol $\mathcal{V}$, and witnessing template $\mathbb{J}$ that achieves the following. Given a dataset $D^*$ and hyperparameters $M^*$, an honest Prover uses $\mathcal{P}$ to execute a training run and get $(W^*, J^*) = \mathcal{P}(D^*, M^*, c_1)$, where $W^* \in \mathbb{R}^d$ is a final weight vector, $J^* \in \mathbb{J}$ is a witness to the computation, and $c_1 \sim C_1$ is an irreducible source of noise. The Verifier must accept this true witness and resulting set of model weights with high probability: $\mathrm{Pr}_{c_1 \sim C_1, c_2 \sim C_2}[\mathcal{V}(D^*, M^*, J^*, W^*, c_2) = 1] \geq 1 - \delta_1$ , where $\delta_1 \ll 1/2$ and $c_2$ is the randomness used by the Verifier.*

*Conversely, $\forall$ computationally-feasible probabilistic adversaries $\mathcal{A}$ which produce spoofs $(D, M, J) = \mathcal{A}(D^*, M^*, J^*, W^*, c_3)$ where $D \neq D^*$ and $c_3 \sim C_3$ is the randomness used by the adversary, the Verifier must reject all such spoofs with high probability: $\mathrm{Pr}_{c_1 \sim C_1, c_2 \sim C_2, c_3 \sim C_3}[\mathcal{V}(D, M, J, W^*, c_2) = 0] \geq 1 - \delta_2$ where $\delta_2 \ll 1/2$.*

Following the literature on the related Proof-of-Learning problem [JYCC+21], we use as a witness the series of $m$ model weight checkpoints $J^* = \mathcal{W} = (W_0, W_1, \ldots, W_{m-1}, W^*)$. Model weight checkpoints are already routinely saved throughout large training runs; we assume a checkpoint is saved after training on each $k = n/m$-datapoint segment. During verification, the Prover provides the Verifier with the *training transcript* $T = \{D, M, \mathcal{W}\}$, which the Verifier will then test to check its truthfulness.

In practice, we cannot yet pursue provable robustness to *all* probabilistic adversaries $\mathcal{A}$, due to the nascent state of the theoretical foundations of deep learning [FJT+22]. Instead, as is done in the PoL literature, we approximate robustness to $\mathcal{A}$ by proposing a range of adversaries and then showing a protocol that defeats them. In particular, we will consider two major types of attacks (and their combinations):

- *Non-Uniqueness Attack*: The Prover forges a different $D \neq D^*$ and $M$ (and corresponding checkpoints $\mathcal{W}$) that would, if training was reexecuted, also lead to $W^*$. In general, it is easy to produce such transcripts (e.g., initialize training at $W_0 = W^*$ and take no steps). In Section 4.3 we propose constraints on $\mathcal{P}$ that block such attacks.

- *Data Subtraction Attack*: The Prover reports training on $D$, but secretly only trains on a subset $D^* \subsetneq D$.

- *Data Addition Attack*: The Prover reports training on $D$, but secretly trains on an additional set of data $D^* \supsetneq D$.

- *Checkpoint Glue-ing Attack*: The Prover reports a pair of successive checkpoints $(W_i, W_{i+1}) \subset \mathcal{W}$, but $W_{i+1}$ was not produced from $W_i$ by a training procedure at all.

We address the latter three attacks in Section 6.

As a brute-force solution to Proof-of-Training-Data, the Verifier could simply re-execute the complete training process defined by $T$, and check that the result matches $W^*$. However, beyond technical complications[2], doing so is far too computationally expensive to be done often; a government Verifier would need to be spending as much on compute for audits as every AI developer combined. Therefore any verification protocol $\mathcal{V}$ must also be *efficient*, costing much less than the original training run. Inevitably, such efficiency makes it near certain that the Verifier will fail to catch spoofs $D \neq D^*$ if $D$ only differs in a few data points; in practice, we prioritize catching spoofs which deviate on a

---

[2]This would also fail in practice because of irreducible hardware-level noise which means that no two training runs return exactly the same final weight vector [JYCC+21]. a transcript could still be examined piecewise, as done in [JYCC+21]; for more, see Section 4.1.

substantial fraction of points in $D^*$. Though we do not restrict to a particular definition of dataset deviations, we list several possibilities relevant for different Verifier objectives in Appendix D.

## 4   Verification Strategies

We provide several complementary tools for detecting whether a transcript $T$ is spoofed. Combined, these methods address many different types of attacks, including all current attacks from the PoL literature [ZLD$^+$22, FJT$^+$22].

### 4.1   Existing Tools from Proof-of-Learning

Our protocol will include several existing spoof-detection tools from the Proof-of-Learning literature [JYCC$^+$21], such as looking for outliers in the trajectory of validation loss throughout training, and plotting the segment-wise weight-change $\|W_i - W_{i-1}\|_2$ between the checkpoints $\mathcal{W}$. The most important of these existing tools is the segment-wise retraining protocol of [FJT$^+$22]. Let $R(W_{i-1}, \Pi_i, c; M)$ be the model training operator that takes in a weight checkpoint $W_{i-1}$, updates it with a series of gradient steps based on training data sequence $\Pi_i$ (describing the order in which the Prover claims data points were used in training between checkpoints $W_{i-1}$ and $W_i$, which may be different from the order of the dataset $D^*$), hyperparameters $M$, and hardware-noise-randomness $c \sim C$, and then outputs the resulting weight checkpoint $W_i$. Transcript segment $i$ is $(\epsilon, \delta)$-*reproducible* if for the pair of checkpoints $(W_{i-1}, W_i)$ in $\mathcal{W}$, the reproduction error (normalized by the overall segment displacement) is small:

$$\Pr_{c \sim C} \left( \frac{\|\hat{W}_i - W_i\|_2}{\frac{\|\hat{W}_i - W_{i-1}\|_2 + \|W_i - W_{i-1}\|_2}{2}} < \epsilon \right) > 1 - \delta \quad \text{where} \quad \hat{W}_i = R(W_{i-1}, \Pi_i, c; M). \quad (1)$$

The values $\epsilon$ and $\delta$ trade off false-positive vs. false-negative rates; see [JYCC$^+$21, FJT$^+$22] for discussion. The Verifier can use this retraining procedure as a ground-truth for verifying the faithfulness of a suspicious training segment. However, this test is computationally-intensive, and can thus only be done for a small subset of training segments. Our other verification strategies described in Sections 4.2 and 4.3 will be efficient enough to be executable on every training segment.

### 4.2   Memorization-Based Tests

The simplest way for a Prover to construct a spoofed transcript ending in $W^*$ is to simply make up checkpoints rather than training on $D^*$, and hope that the Verifier lacks the budget to retrain a sufficient number of checkpoints to catch these spoofed checkpoints. To address this, we demonstrate a heuristic for catching spoofed checkpoints using a small amount of data, based on what is to the best of our knowledge a previously-undocumented phenomenon about local training data memorization.

Machine learning methods notoriously overfit to their training data $D$, relative to their validation data $D_v$. We can quantify the degree of overfitting to a single data point $d$ on a loss metric $\mathcal{L}$ : $\mathbb{X} \times \mathbb{R}^{|W|} \to \mathbb{R}$ relative to a validation set $D_v$ via a simple memorization heuristic $\mathcal{M}$:

$$\mathcal{M}(d, W) = \mathbb{E}_{d' \in D_v}[\mathcal{L}(d', W)] - \mathcal{L}(d, W). \quad (2)$$

Recall that $\Pi_i$ is the sequence of data points corresponding to the $i$th segment of the training run. One would expect that in checkpoints before data segment $i$, for data points $d \in \Pi_i$, memorization $\mathcal{M}(d, W_{j<i})$ would in expectation be similar to the validation-set memorization; after data-segment $i$, one would expect to see higher degrees of overfitting and therefore $\mathcal{M}(d, W_{j>i})$ would be substantially higher. We find evidence for this effect in experiments on GPT-2-Small [RWC$^+$19] and the Pythia suite [BSA$^+$23]). As shown in Figures 1 and 2, when a Prover reports the true training data, on average the greatest memorization occurs where $\Pi_i$ and $W_{j=i}$ match. We corroborate this finding with additional experiments on a range of models in Appendix G. The finding is even clearer if we look at jumps in memorization level, which we call the Memorization Delta $\Delta_{\mathcal{M}}$:

$$\Delta_{\mathcal{M}}(d, i; \mathcal{W}, D_v, \mathcal{L}) = \mathcal{M}(d, W_i) - \mathcal{M}(d, W_{i-1}). \quad (3)$$

To test whether each reported checkpoint $W_i$ resulted from training on at least some of the segment training data $\Pi_i$, a Verifier can compute a memorization plot like the one shown in Figure 1. Such



Figure 1: Plots from a GPT-2 experiment, demonstrating the local memorization effect. The maximum score for each data segment (row) is marked with a red box. The largest average memorization for data sequence $\Pi_i$ occurs at the immediately-subsequent checkpoint $W_i$. From left to right: plots of the average loss $\mathcal{L}$; memorization $\mathcal{M}$; and memorization-delta $\Delta_{\mathcal{M}}$; along with the average memorization over time for each segment $\Pi_i$, recentered such that $W_i$ is at $x = 0$.

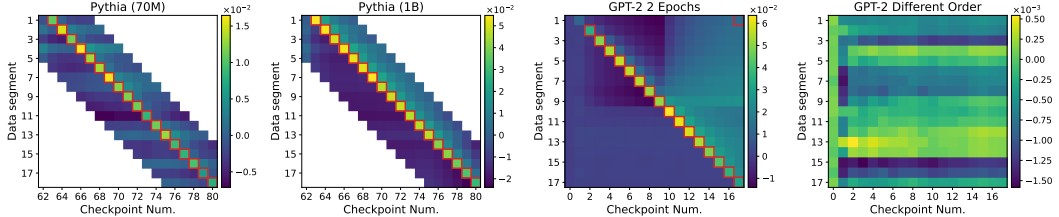

Figure 2: Plots of the memorization $\mathcal{M}$ on other types of training runs, similar to Figure 1. For efficiency, plots of the Pythia models use only $10\%$ of training data, and look only at a window of checkpoints around $W_i$. From left to right: $\mathcal{M}$ for checkpoints near the middle of the Pythia (70M) training run shows the same pattern as GPT-2; checkpoints near the middle of the Pythia (1B) training run show that the phenomenon gets clearer as the model size increases; $\mathcal{M}$ for a GPT-2 run with two epochs over the same data (with random data order each epoch; the first epoch ends at checkpoint 9) to demonstrate that the effect is present over multiple epochs; $\mathcal{M}$ for a GPT-2 run using a random data order other than $\Pi$ shows that the effect is tied to the training data sequence itself.

plots can be computed more efficiently by sampling only a small fraction $\alpha$ of the training data $\Pi$, and by plotting only a few checkpoints $W_{i-\beta}, \dots, W_{i+\beta}$ for each segment $\Pi_i$.

We can further harness this memorization phenomenon to test whether on segment $i$, rather than training on the full claimed data sequence $\Pi_i$ and yielding $W_i$, the Prover secretly skipped training on *at least* a $\kappa$-fraction of the points in $\Pi_i$ and yielded $W_i'$. Consider the odds that, for $d \sim \Pi_i$, $\Delta_{\mathcal{M}}(d, W_i)$ happens to fall in the bottom $p$-probability quantile of the validation set $D_v$'s $\Delta_{\mathcal{M}}$ values on $W_i$:

$$\mathrm{PBQ}(d, p, W_i) = \mathbb{I}\left(\mathbb{E}_{d' \sim D_v} \mathbb{I}(\Delta_{\mathcal{M}}(d', W_i) > \Delta_{\mathcal{M}}(d, W_i)) \leq p\right) \tag{4}$$

$$\mathrm{FBQ}(\Pi, p, W_i) = \mathbb{E}_{d \sim \Pi}\left[\mathrm{PBQ}(d, p, W_i)\right], \tag{5}$$

where $\mathbb{I}$ is the indicator function, PBQ stands for "Point is Below Quantile", and FBQ stands for "Fraction Below Quantile". We can see in Figure 3 that, as expected, when the points in $\Pi_i$ are all included in training, $\mathrm{FBQ}(\Pi_i, p, W_i)$ is small compared to $\mathrm{FBQ}(\Pi_{j \neq i}, p, W_i)$). If many points were secretly excluded, as in $W_i'$, we should expect that $\mathrm{FBQ}(\Pi_i, p, W_i')$ should be higher and closer to $\mathrm{FBQ}(D_v, p, W_i')$, where $D_v$ is the validation set. If the Prover secretly excluded a greater than $\kappa$-fraction of data points in $\Pi_i$ thus yielding $W_i'$, then we should expect that:

$$\mathrm{FBQ}(\Pi_i, p, W_i') = (1 - \kappa)\mathrm{FBQ}(\Pi_i, p, W_i) + \kappa\mathrm{FBQ}(D_v, p, W_i') \geq \kappa \cdot \mathrm{FBQ}(D_v, p, W_i') \tag{6}$$

(see derivation in Appendix B). Rearranging terms, we get

$$\lambda(\Pi_i, p, W_i') := \frac{\mathrm{FBQ}(\Pi_i, p, W_i')}{\mathrm{FBQ}(D_v, p, W_i')} \geq \kappa. \tag{7}$$

$\lambda(\Pi_i, p, W_i')$ can be estimated using a small fraction of training and validation datapoints, and can serve as an upper-bound estimate on $\kappa$, the fraction of $\Pi_i$ secretly excluded from training $W_i'$.[3] In

---

[3]The percentile-threshold $p$ is left unspecified, but should be kept $\ll 0.5$. The test can be strengthened by varying the chosen fraction $p$ and rerunning the analysis to confirm its insensitivity.

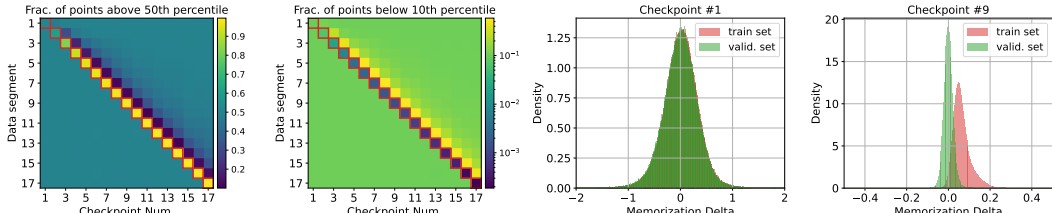

Figure 3: Exploring the pointwise memorization effect on GPT-2. From left to right: For each $W_i$ and $\Pi_i$, we plot the fraction of points with $\Delta_{\mathcal{M}}$ above the median $1 - \text{FBQ}(\Pi_i, 0.5, W_i)$, and see that in the diagonal segments, most individual points are above the median. This shows that memorization occurs pointwise, suggesting that it can be detected via sparse random sampling. The highest segment in each row is surrounded by a red box; Plotting the fraction of samples *below* the 10%ile, we see the fraction is uniquely low on diagonal tiles $(\Pi_i, W_{j=i})$, as predicted; two histograms comparing $\Delta_{\mathcal{M}}$ for diagonal vs. nondiagonal weight checkpoints across all data segments, shows how the distributions may or may not overlap. Even at Checkpoint 1, the leftmost plot shows that $\Pi_1$ has a (marginally) larger fraction of points above the median than any other data segment.

Section 6 we show that this heuristic can detect even small data subtractions in practice, and in Appendix H.2 we show the test's effectiveness across a range of percentiles $p$ and segment-lengths $k$.

We also observe that $\mathcal{M}$ gradually decreases over time from an initial peak immediately after the point's training segment. This echoes the many findings on "forgetting" in deep learning [TSC+18]. We show in Section 6 how this can be used to catch gluing attacks.

### 4.3 Fixing the Initialization and Data Order

As mentioned in Section 3, a Proof-of-Training-Data protocol needs to defend against non-uniqueness attacks, by making it difficult for a malicious Prover to produce a second transcript with $D \neq D^*$ that, if training was legitimately executed, would *also* end in $W^*$. There are two well-known types of attacks the Prover might use to efficiently produce such spoofs:

- *Initialization attacks*: An attacker can choose a "random" initialization that places $W_0$ in a convenient position, such as close to the target $W^*$. Even if the Verifier uses statistical checks to confirm that the initialization appears random, these are sufficiently loose that an adversary can still exploit the choice of initialization [ZLD+22].

- *Synthetic data/data reordering attacks*: Given the current weight vector $W_i$, an attacker can synthesize a batch of training datapoints such that the resulting gradient update moves in a direction of the attacker's choosing, such as towards $W^*$. This could be done through the addition of adversarial noise to existing data points [ZLD+22], generating a new dataset [TJSP22], or by carefully reordering existing data points in a "reordering attack" [SSK+21].

We propose methods for preventing both of these attacks by forcing the Prover to use a certified-random weight initialization, and a certified-random data ordering. The randomized data ordering guarantees that the adversarial Prover cannot construct synthetic datapoints that induce a particular gradient, because it does not know the corresponding weights $W$ at the time of choosing the datapoints $D$. Given a fixed data ordering, we discuss in Appendix E why it may be super-polynomially hard to find a certified-random weight initialization that, when fully trained, results in a particular $W^*$.

The Verifier can produce this guaranteed-random initialization and data order by requiring the Prover to use a particular random seed $s$, *constructed as a function of the dataset $D$ itself*. This produces the initialization $W_0 = G_r(s) \in \mathbb{X}^n$ and data ordering $S = G_p(s)$ using a publicly known pseudorandom generators $G_r$ and $G_p$.[4][5] The Prover can also construct a verifiable validation subset $D_v$ by holding

---

[4]$G_r$ is a cryptographically-secure pseudorandom $d$-length vector generator, with postprocessing defined in the hyperparameters $M$, and $G_p$ is a publicly-agreed pseudorandom $n$-length permutation generator. $G_p$ can be modified to repeat data multiple times to train for multiple epochs, or according to a randomized curriculum.

[5]In practice, the statistical test to verify that the certified ordering was used will only be able to distinguish whether each data point $d_i \sim D$ was trained in the assigned segment $S_i$ or not. Therefore, for this protocol to apply a checkpoint must be saved at least twice per epoch, $k \leq n/2$.

out the last $n_v$ data-points in the permutation $S$ from training. The Prover constructs $s$ as follows. Assume that the dataset $D$ has some initial ordering. Let $H$ be a publicly-known cryptographic hash function. We model $H$ as a random oracle, so that when composed with $G_r$ or $G_p$, the result is polynomial-time indistinguishable from a random oracle.[6] This means that if a Prover wants to find two different seeds $s_1, s_2$ that result in similar initializations $W_{0;1}, W_{0;2}$ or two similar permutations $S_1, S_2$, they can find these by no more efficient method than guessing-and-checking. For large $d$ and $n$, finding two nontrivially-related random generations takes exponential time. We construct the dataset-dependent random seed $s$ as

$$s(D, s_{rand}) = H\left(H(d_1) \circ H(d_2) \circ \cdots \circ H(d_a) \circ s_{rand}\right), \tag{8}$$

where $\{d_1, \ldots, d_a\} = D$, $\circ$ is the concatenation operator, and $s_{rand}$ is a Prover-chosen 32-bit random number to allow the Prover to run multiple experiments with different seeds.[7] A Verifier given access to $D$ (or only even just the hashes of $D$) can later rederive the above seed and, using the pseudorandom generators, check that it produces the reported $W_0$ and $S$.

The important element of this scheme is that given an initial dataset $D^*$ and resulting data order $S$, modifying even a single bit of a single data point in $D^*$ to yield a second $D$ will result in a completely different data order $S'$ that appears random relative to $S$. Thus, if we can statistically check that a sequence of checkpoints $\mathcal{W}$ matches a data order $S^*$ and dataset $D^*$ better than a random ordering, this implies that $D^*$ is the *only* efficiently-discoverable dataset that, when truthfully trained[8], would result in the checkpoints $\mathcal{W}$ and final weights $W^*$. We provide this statistical test in Appendix C.

This same approach can be extended to the batch-online setting, where a Prover gets a sequence of datasets $D_1^*, D_2^*, \ldots$ and trains on each before seeing the next. The Prover simply constructs a new seed $s(D_i^*, s_{rand})$ for each dataset $D_i^*$, and continues training using the resulting data ordering. This works so long as each $D_i^*$ is large enough for a particular data-ordering to not be brute-forceable.

### 4.4 Putting It All Together

In Appendix A we sketch a complete protocol for combining these defenses complementarily to detect all of the attacks discussed in Section 6. The overall computational cost for the Verifier is $O(n)$ training data-point hashes, $O(\alpha n)$ model inferences for computing losses, and $O(|Q|n)$ gradient computations for retraining transcript segments (where $|Q|$ depends on hyperparameters that can be adjusted according on the Verifier's compute budget). Importantly, the Verifier's cost grows no worse than linearly with the cost of the original training run. If we run our tests using an $\alpha = 0.01$ fraction of the points in each segment as done in our experiments below, the verification cost of computing our new tests in Sections 4.2 and 4.3 totals just 1.3% of the original cost of training, assuming inference is $3\times$ cheaper than training.

## 5 Experimental Setup

Our main experiments are run on GPT-2 [RWC+19] with 124M parameters and trained on the OpenWebText dataset [GCPT19]. We use a batch size of 491,520 tokens and train for 18,000 steps (~8.8B tokens), which is just under 1 epoch of training, saving a checkpoint every 1000 steps. See Appendix F for additional details. The data addition attack experiments in Section 6 further use the Github component of the Pile dataset [GBB+20] as a proxy for a Prover including additional data that is different from reported data. In addition to training our own models, we also evaluate Pythia checkpoints [BSA+23] published by EleutherAI, as they publish the exact data order used to train their models. We chose the 70M, 410M, and 1B-sized Pythia models trained on the Pile dataset with deduplication applied. All experiments were done using 4 NVIDIA A40 GPUs.

---

[6]Since the random oracle model is known to be unachievable in practice, we leave the task of finding a more appropriate cryptographic primitive as an interesting direction for future work.

[7]To enable a Prover to only reveal the required subset of data to the Verifier, it may be best to construct $s$ using a Merkle hash tree.

[8]It is still possible to construct multiple data sets $D_1, D_2$, and train on both, interleaving batches. This is not a uniqueness attack, but a data addition attack, and will be addressed in Section 6.

# 6 Empirical Attacks and Defenses

Below, we show that our methods address the last three attacks (Checkpoint Glue-ing, Data Subtraction, and Data Addition). We omit the synthetic initialization and synthetic data attacks of [FJT$^+$22, ZLD$^+$22] as we addressed those in Section 4.3. All plots are from experiments using GPT-2; we include additional experiments in Appendix H. We do not claim that the attacks studied here are exhaustive, but provide them as a starting point to motivate future work.

**Checkpoint Glue-ing Attack** A known attack against Proof-of-Learning, which also applies to PoTD, is to "glue" two training runs $W^A$ and $W^B$ together and report a combined sequence of checkpoints $\mathcal{W} = (W_0^A, \ldots, W_i^A, W_{j\gg0}^B, \ldots, W_{final}^B)$. The resulting model $W_{final}^B$ can be trained on undisclosed data prior to segment $j$, with the Prover never reporting this data to the Verifier. As highlighted by [JYCC$^+$21], the size of the glued segment $\|W_j^B - W_i^A\|_2$ will generally appear as an outlier in weight-space. We demonstrate this phenomenon in Figure 4. Following [JYCC$^+$21], a Verifier could then check such suspicious segments via retraining. We demonstrate a second verification option using inference instead of training: the Verifier can check whether the checkpoint $W_j^B$ has memorized not only the most recent data $\Pi_i$, but also the preceding data segments $\Pi_{i-1}, \Pi_{i-2}, \ldots$ The absence of long-term memorization is visible in the memorization heatmap in Figure 4.

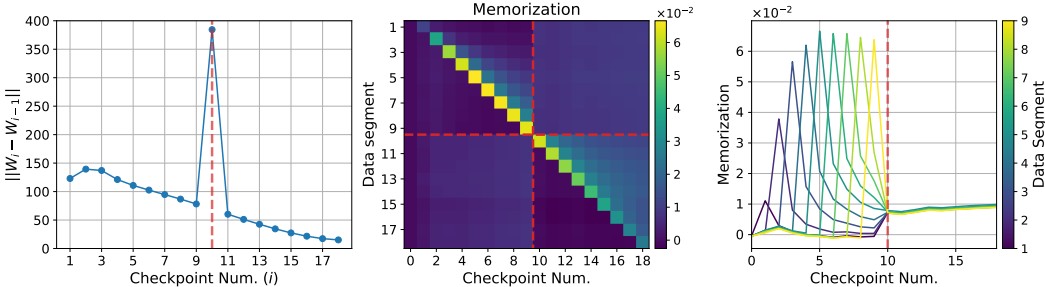

Figure 4: Exploring how the defenses handle a simulated gluing attack, where the transcript switches from one GPT-2 training run to a second run after the 9th checkpoint. (We assume the Prover uses a certified data order (Section 4.3) on the sections before and after gluing.) From left to right: The norm of the weight-changes jumps abruptly during the gluing, causing the Verifier to flag that checkpoint as suspicious.; the Verifier creates a memorization plot (shown here with 100% sampling rate for clarity), and discovers the gluing by spotting that memorization of past checkpoints cuts off abruptly at the suspicious segment.; The same long-term memorization cutoff effect is visible plotting average $\mathcal{M}$ for each data segment across time.

To avoid the spike in weight-space shown in Figure 5 when jumping from $W_i^A$ to $W_j^B$, the attacker can break up the large weight-space jump into smaller jumps by artificially constructing intermediate checkpoints $aW_j^B + (1-a)W_i^A$ for several values of $a$. However, these interpolated checkpoints fail our memorization tests, as they are artificial and not the result of actual training (Figure 5).

**Data Subtraction Attack** In a data subtraction attack, a Prover claims the model has been trained on more points than it truly has. Detecting data subtraction attacks could enable a Verifier to detect overclaiming by model providers, including claiming to have included safety-related data when they secretly did not. Subtraction can also be used to hide data addition attacks, as combining the two attacks would mean the segment was still trained on the correct number of datapoints, thus suppressing the weight-change-plot signature used to catch data addition (as in Figure 7). We demonstrate the effectiveness of an efficient memorization-based approach for detecting subtraction, described in Section 4.2. Leveraging the subtraction-upper-bound test from Equation 7, we see in Figure 6 that the upper-bound heuristic $\lambda(\Pi, p, W_i)$ is surprisingly tight, consistently differentiating between no-subtraction segments and even small subtraction attacks. Still, even if $\lambda(\Pi_i, p, \mathcal{W}) > z$ for some large $z \gg 0$, this is only an upper bound on the quantity of data subtraction, and does not prove that a $z$-fraction of points were subtracted. The Verifier can instead use this test as an indicator to flag segments for retraining, which would confirm a subtraction attack. (That retraining would

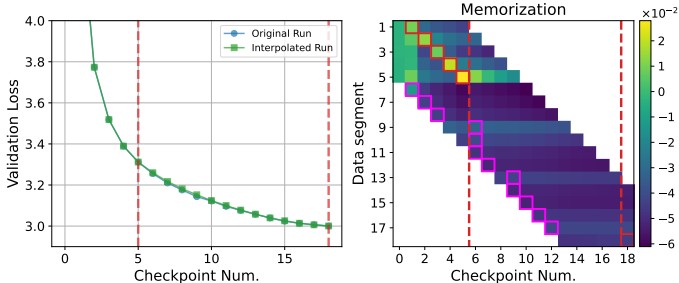

Figure 5: Simulating an interpolation attack by training a GPT-2 model until the 5th checkpoint, and then linearly-interpolating to a final checkpoint. On the left, we show that an attacker can carefully choose interpolation points to mask any irregularities in validation loss. (The green line perfectly overlaps with the blue line.) Nonetheless, on the right, we see a clear signature in the memorization plot, computed using only 1% of data: the typical memorization pattern along the diagonal does not exist for the interpolated checkpoints. For each row corresponding to a data segment $\Pi_i$, a box marks the maximal-$\mathcal{M}$ checkpoint. The box is red if the checkpoint is a match $W_i$, and magenta if there is no match and the test fails $W_{j \neq i}$.

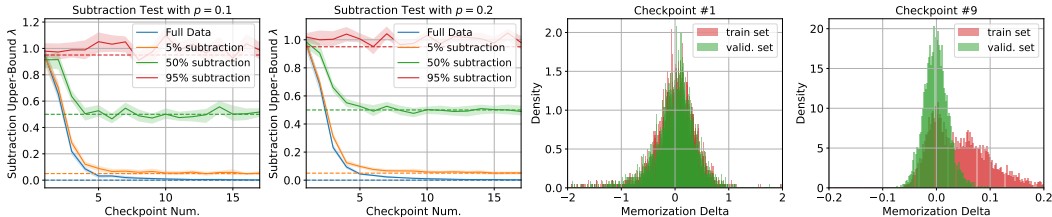

Figure 6: On the left, we simulate a data subtraction attack with different levels of subtraction (0%, 5%, 50%, or 95%) in a GPT-2 training run. The plots show the results of computing the subtraction-upper-bound heuristic $\lambda(\Pi_i, p, W_i)$ for each checkpoint, using just 1% of training data, across 20 random seeds, with dashed lines showing the true subtraction rate. $\lambda$ estimates the maximum level of data subtraction in each segment. We see that $\lambda$ provides a surprisingly tight upper bound for the honestly-trained segment, while providing no such upper bound for the larger subtraction attacks. To illustrate the logic behind this test, on the right, we show how a 50% subtraction attack can create a bimodal distribution of $\Delta_{\mathcal{M}}$ values. $\lambda$ captures the relative weight of the left mode.

result in a different weight vector can be inferred from the plot of the 50%-addition attack in Figure 7). Appendix H.2 explores the test's performance on the suite of Pythia models.

**Data Addition Attack**    Addition attacks occur when a Prover in addition to training on the declared dataset $D$ and data sequence $\Pi$, trains on additional data $D'$ without reporting it to the Verifier. This can be used to secretly enhance a model's capabilities, install backdoors, or train on restricted data. This attack cannot be detected using memorization analysis (Figure 7), because the Verifier does not know and cannot test points $d' \in D'$. However, Figure 7 also shows that significant amount of data addition, or data addition from a different distribution ,causes a larger weight-displacement in that segment, which can be detected by looking at the segment's magnitude. If a Prover tries to hide this by deleting an equally-sized subset of points from $D$, that can be detected as a subtraction attack. We leave exploration of other attacks and defenses, like changing learning rates, to future work.

This raises the problem of how to estimate the "expected" segment length, which may require retraining on a chosen subset of segments, and interpolating to estimate other segments' lengths. If, to hide the non-smoothness of adding extra data to a single segment, the Prover adds data uniformly throughout a large fraction of the training run, then choosing even a small number of segments randomly should be sufficient to catch at least one offending segment with high probability. Unfortunately, these defenses would not detect an attacker that adds a modest amount of data within a small number of segments, that cannot be detected by the segment-magnitude or data subtraction defenses.

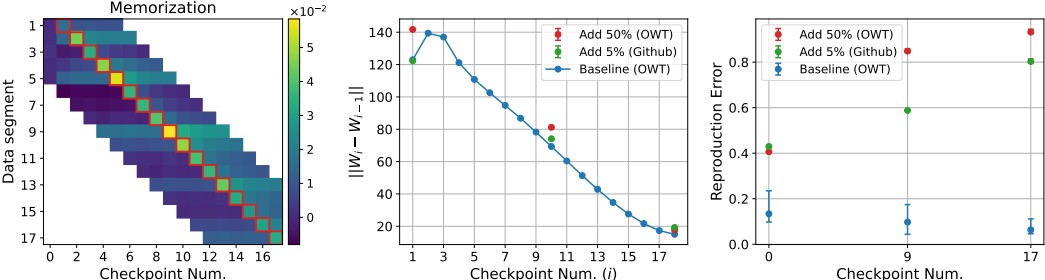

Figure 7: Simulating a data addition attack by picking a single segment (1st, 10th, or 18th), and adding 50% more data from the same distribution (OpenWebText), or 5% more data from a different distribution (Github), or no data addition (truthful reporting). Results are shown with error bars across 4 random seeds. From left to right: the memorization test with $1\%$ of samples does not spot any differences even with 50% data addition; Plotting weight-changes between checkpoints, a Verifier can see a suspicious spike at the attacked segments; The Verifier retrains the suspicious segments and checks the distance between the reported and re-executed checkpoint weights.

# 7    Discussion and Limitations

This work contributes to an emerging societal effort to develop practical and robust tools for accountability in the large-scale development of AI models. The statistical tests we introduce are best taken as an opening proposal. Future work could propose clever new attacks that break this protocol, or better yet, create new defenses that efficiently detect more, and subtler, attacks and enable trustworthy verification of ML models' training data.

**Experimental Limitations**    This work provides suggestive evidence for the local-memorization phenomenon, but further study is needed across additional modalities, architectures, and training recipes in order to determine its broad applicability. Encouragingly, we find in Appendix G that local-memorization gets even stronger as models get larger, though memorization appears weaker near the end of training as the learning rate shrinks. The paper's experiments only include language models, in part because they are a current priority for audits. The memorization tests used may need to be adjusted for models trained with less data on many epochs, such as image models [YZS⁺22].

**Attacks Our Protocol Does Not Catch**    There are several remaining directions for attacks. The attacks explored above can be composed in new ways, and it may be possible for compositions of attacks to undermine the defenses that would otherwise detect each attack individually. The method also does not address small-scale data additions, and thus cannot yet detect copyright violations or spot inserted backdoors [XWL⁺21]. It also cannot detect attacks based on small-norm modifications to the weights, which could be used to insert backdoors [BISZ⁺22]. Finally, attacks could be masked with cleverly chosen hyperparameters, such as by using a temporary lower-than-reported learning rate to shrink large changes in $W$. Exploring whether such attacks are feasible without degrading learning performance – and identifying defenses – is an interesting direction for future work.

**Applicability to Different Training Procedures**    We attempted to make our procedure as agnostic as possible to the details of the training procedure, and believe it will be compatible with most training procedures for large models in use today. However, our protocol does not apply to online or reinforcement learning, or to schemes that require multiple models to be co-trained [GPAM⁺20], as the data is unknown in advance. This means the uniqueness defense cannot be applied (Section 4.3). Finding methods for defending against non-uniqueness attacks even in the online setting is a valuable direction for future work.

**Maintaining Privacy and Confidentiality**    One significant challenge to using this protocol in practice is that, just like with PoL [JYCC⁺21], it requires that the Prover disclose confidential information to the Verifier, including training data, model weights, and code. In principle, the Prover may only need to disclose hashes of the data and weights to the Verifier, with the matching full data and weights only ever supplied on the secure cluster during verification.

## Acknowledgements

We thank Nicolas Papernot, Anvith Thudi, Jacob Austin, Cynthia Dwork, Suhas Vijaykumar, Rachel Cummings Shavit, Shafi Goldwasser, Hailey Schoelkopf, Keiran Paster, Ariel Procaccia, and Edouard Harris for helpful discussions. DC was supported by NSERC CGS-D, and DC and YS are supported by Open Philanthropy AI Fellowships.

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

# A    Combined Verification Protocol

We can unify the defenses of Section 4 into a combined defense protocol, which catches a wide swath of attacks, including all current attacks on from the Proof-of-Learning literature [FJT$^+$22, ZLD$^+$22].

A Prover gives the Verifier a transcript $T = \{D, M, \mathcal{W}\}$ and a final weight vector $W^*$. The verifier proceeds to verify whether $T$ is a valid training transcript through the following checks:

1. Check that $\mathcal{W}$ ends in the claimed final weights $W^*$.

2. Given the dataset $D$, hash it to yield the seed $s$ as in Section 4.3, and use that seed compute the resulting data order $\Pi$ and validation subset $D_v$. (Alternatively, these hashes can be provided by the Prover, and only verified when each point is needed for the protocol.)

3. Check that $W_0$ matches $G_r(s)$. If this fails, reject the transcript.

4. Create an empty list $Q$ to store all suspicious-looking segments to retrain. For each segment $W_i, \Pi_i$, include it in the list $Q$ to retrain if it fails any of the following checks:

    (a) Randomly select an $\alpha$ fraction (e.g., 1% of $k$) of points $\Pi_{i,\alpha}$ from $\Pi_i$. For each such point $d \sim \Pi_{i,\alpha}$, compute the losses on $W_i$ and $W_{i-1}$, shorthanded as sets $\mathcal{L}_{\Pi_i,i}$ and $\mathcal{L}_{\Pi_i,i-1}$. Similarly, pick an $\alpha$ fraction [9] of points from the validation set $D_v$ and compute these points' losses on $W_i$, shorthanded as $\mathcal{L}_{D_v,i}$. (The validation loss on $W_{i-1}$, $\mathcal{L}_{D_v,i-1}$, should've already been computed when looping on the previous segment.) Also, randomly select an $\alpha k$ subset of data points $D_t$ from across all training segments $\Pi$, and compute these points' losses on $W_i$, $\mathcal{L}_{D_t,i}$.

        - If the Verifier wants to plot complete memorization plots, for example as a sanity check or to use in checking for a Glue-ing attack as described in Section 6, they can also compute the losses on $2\beta$ nearby weight checkpoints $W_{i-\beta}, \ldots, W_{i+\beta-1}$. However, this is not part of the core protocol, and will not be counted in its sample complexity.

    (b) Compare the values in $\mathcal{L}_{\Pi_i,i}$ and $\mathcal{L}_{D_t,i}$ using the one-sided binomial hypothesis test described in Appendix C, to check that the model used the correct data ordering. If the test cannot reject the null hypothesis that $\Pi_i$ does not match $W_i$ better than randomly-selected training points $D_t$, then run the test on an even larger fraction of data points from $\Pi_i$ and $D$ to double-check the statistical significance of the null. If this still fails, it suggests that the transcript may have been spoofed. Investigate further (such as by reexecuting training on this segment) and/or reject the transcript.

    (c) Using $\mathcal{L}_{\Pi_i,i}, \mathcal{L}_{\Pi_i,i-1}, \mathcal{L}_{D_v,i}$, and $\mathcal{L}_{D_v,i-1}$, calculate a data subtraction upper bound heuristic as in Section 4.2. If this heuristic fails to bound the data subtraction to below an acceptable small limit (which depends on how strongly the Verifier wants to catch even small subtraction attacks, and on the Verifier's retraining budget), add this segment to $Q$.

    (d) Using $\{\mathcal{L}_{D_v,j} \mid j \in 0, \ldots, m\}$, compute the mean validation set loss across time, and check that it is smooth at $W_i$ and doesn't increase or decrease discontinuously. If it does, add this segment to $Q$. (The tolerance should depend on the retraining budget.)

    (e) Compute the distance $\|W_i - W_{i-1}\|$ from the previous checkpoint. Check that this delta is similar to nearby segments' weight-space deltas. If it is not, add this segment to $Q$. (The tolerance should depend on the retraining budget. We leave the question of how best to set this threshold, and of $\sigma$ below, to minimize false negatives while avoiding reducing false positives that increase retraining costs, to future work.)

5. Randomly sample $\sigma$ additional data segments from throughout training, and add them to $Q$. These additional segments are important to establish baseline estimates of segments' weight-space deltas across training, to ensure that there were no systematic data addition attacks at every segment. (Illegal data additions in every segment would shift the entire weight-change delta magnitude plot, thus suppressing anomalies in any one segment).

6. For each segment in the list $Q$, execute retraining and verify that the resulting weights $\hat{W}_i$ are within an $\epsilon$-ball of the original reported weights $W_i$ reported in the transcript.

---

[9] To reduce noise when comparing validation performance across checkpoints, this $\alpha$ subset of $D_v$ should be the same across all evaluated checkpoints.

If any values in the re-trained weights fail to come within the tolerance $\epsilon$, that is significant evidence that the transcript has been spoofed, and warrants further investigation. For example, the segment can be retrained more times, to confirm that the weight-space variance across retraining results $\hat{W}^{(1)}, \hat{W}^{(2)}, \ldots$ is sufficiently smaller than $\epsilon$ such that the reported $W_i$ is a clear outlier.

If all these tests pass, accept the transcript.

### A.1  Complexity

The time costs of training, borne by the the Prover are:

1. $h \times |D|$, where $h$ is the cost of a hash, for generating the initial random seed.
2. $s \times n$, where $s$ is the cost of a single gradient computation, and $n$ is the number of training data points.

In comparison, the time costs to the Verifier (assuming the transcript is accepted) are:

1. $h \times |D|$ hashes for verifying the initial weights.
2. $(2 + 1 + 1) \times \alpha \times \frac{s}{3} \times n$ operations for computing the loss of an $\alpha$ fraction of datapoints in $\Pi_i$ on $W_i$ and $W_{i-1}$, and another $2\alpha$ fraction of points in $D_t$ and $D_v$. We also assume that computing the loss requires $\frac{1}{3}$ the number of operations as computing a gradient update, which is the standard ratio of inference vs. training when using backpropagation.
3. $s \times n \times |Q|/m$ operations for retraining, where $m$ is the total number of checkpoints in the training run.

## B  Subtraction Upper-Bound Heuristic Derivation

Let $W_i$ be the checkpoint obtained by training on the full claimed data sequence $\Pi_i$, and let $W_i'$ be the checkpoint obtained by excluding $\kappa$-fraction of the points in $\Pi_i$. Our goal is to estimate $\text{FBQ}(\Pi_i, p, W_i')$, the fraction of points in $\Pi_i$ that fall in the bottom $p$-probability quantile of $D_v$'s $\Delta_{\mathcal{M}}$ values on $W_i'$.

Let's start with the distribution of $\Delta_{\mathcal{M}}(d, W_i)$, where $d \sim \Pi_i$. We make the following two assumptions:

1. The subtracted $\kappa$-fraction of points are sampled randomly from $\Pi_i$.
2. The $p$-th percentile of the $\Delta_{\mathcal{M}}$ values of $D_v$'s $\Delta_{\mathcal{M}}$ values on $W_i$ is the same as on $W_i'$.

Firstly, since $\kappa$-fraction of points were subtracted, we know that those subtracted points are distributed like the validation set (assumption 1). Therefore, the fraction of points in $\Pi_i$ that lie in our quantile of interest that were *subtracted* is $\kappa\text{FBQ}(D_v, p, W_i')$.

Next, we must consider the points that were not subtracted but lie in the quantile. Using assumption 2, we know that $(1 - \kappa)$ fraction of points that were below the quantile remain in the quantile. Therefore, the fraction of points in $\Pi_i$ that lie in our quantile of interest that were *not subtracted* is $(1 - \kappa)\text{FBQ}(\Pi_i, p, W_i)$.

Putting it all together, we have

$$\text{FBQ}(\Pi_i, p, W_i') = \kappa\text{FBQ}(D_v, p, W_i') + (1 - \kappa)\text{FBQ}(\Pi_i, p, W_i) \tag{9}$$

There are several things to note about this derivation. First of all, regardless of whether assumption 2 is correct, the second term of Equation 9 is $\geq 0$ and therefore, $\lambda$ is still an upper-bound. In practice, the actual fraction of subtracted points that lie in the quantile might not be exactly equal to $\text{FBQ}(D_v, p, W_i') = p$. Therefore, if $(1 - \kappa)\text{FBQ}(\Pi_i, p, W_i)$ is very small and close to zero, the resulting upper-bound can be smaller than $\kappa$, which is the case in Figure 6.

Lastly, when computing FBQ values, we can use the unnormalized $\mathcal{M}$ values to compute $\Delta_{\mathcal{M}}$ as such:

$$\Delta_{\mathcal{M}}(d, i; \mathcal{W}, \mathcal{L}) = \mathcal{L}(d, W_i) - \mathcal{L}(d, W_{i-1}). \tag{10}$$

This does not affect the FBQ values, since the normalization term $\mathbb{E}_{d' \in D_v}[\mathcal{L}(d', W)]$ does not depend on $d$ and therefore and has no effect when evaluating $\Delta_{\mathcal{M}}(d', W_i) > \Delta_{\mathcal{M}}(d, W_i)$. Removing the normalization term could represent a 2x efficiency gain. We kept the normalization term in the main paper because it may still be important in other cases for preserving the meaning of the Memorization Delta $\Delta_{\mathcal{M}}$, which is used to produce memorization-charts. Without the normalization term, it becomes hard to compare $\Delta_{\mathcal{M}}$ across segments, especially in the earlier stages of training when the changes in loss between checkpoints vary quite a bit across segments.

## C   Data Order Statistical Test

We want a statistical test that will tell the Verifier whether, for a given training dataset $D$ and data ordering $S$, which together yield a data sequence $\Pi$, and for a given weight checkpoint $W_i$, the data segment sequence $\Pi_i \in \mathcal{X}^k$ explains the memorization pattern of $W_i$ better than a random data order/sequence $\Pi'_i$ (which we assume is drawn randomly from $D$). In particular, based on results from Section 4.2, we know that datapoints from the most recent training segment $d \in \Pi_i$ tend to have higher memorization delta values $\Delta_{\mathcal{M}}$ than the average point $d' \in D$ from the overall training distribution $D$. Conversely, points from $\Pi'_i$ would have no reliably greater $\Delta_{\mathcal{M}}$ than the rest of $D$.

We will use the following test, where $\Pi^? = \Pi'_i$ is the null hypothesis and the alternative hypothesis is $\Pi^? = \Pi_i$. Let $z = \text{median}_{d \in D}(\Delta_{\mathcal{M}}(d, W_i))$, estimated via a small number of samples from $D$. For Pick $n_t$ datapoints from data sequence $\Pi^?$, and for each data point $d \in \Pi^?$, check if it's $> z$. Under the null hypothesis, the probability that each point passes this check is $0.5$. Let the test statistic be $t$:

$$t(\Pi^?) = \sum_{d_j \sim \Pi^?, j=1,\ldots,n_t} \mathbb{I}(\Delta_{\mathcal{M}}(d_j, W_i) > z) \tag{11}$$

where $\mathbb{I}$ is the indicator function. The value of $t(\Pi'_i)$, the statistic under the null hypothesis, is distributed as a binomial with biased coin probability $c = 1/2$ and $n_t$ samples. However, we expect that $t(\Pi_i)$ is a binomial with a larger $c$. To compute our confidence that $W_i$ was trained using the data order $\Pi_i$, we can use a one-sided binomial hypothesis test, computing a $p$-value as $1 - CDF_{binomial}(c = 1/2, n_t, k < t)$ where $k$ is the value up to which to calculate the CDF. This statistic can be computed jointly across all checkpoints (requiring relatively few samples per checkpoint) to prove that the overall data ordering matches the one defined in Section 4.3.

Note that this test is similar to the "subtraction upper bound" heuristic from Section 4.2, with the key difference being that in this test we compare against the distribution of all training points $D$ (since the counterfactual is a randomly selected subset of training data), whereas the subtraction test compares against points from the validation set $D_v$ (since the counterfactual is that the points are never included in training). As an additional note, this same test can be generalized by replacing the median with a quantile, which may improve sample efficiency depending on the shape of the $\Delta_{\mathcal{M}}$ distribution on $\Pi_i$ vs. $\Pi'_i$.

# D  Verifier Objectives Table

| Transcript Use-Case | Attacker Motivation | Definition of Defender Success |
|---|---|---|
| Check whether a model $W$ was trained on data from a disallowed distribution (e.g., relating to backdoors, cyberexploit generation, or enabling an undisclosed modality such as images). | A Prover wants to claim that $W$ lacks a certain ability in order to avoid scrutiny, and does so by claiming $W$ has only been trained on data from distribution $\mathcal{D}$ and not on distribution $\mathcal{D}'$. | A test such that, given target weight checkpoint $W$, confirms that its training data did not include, in addition to a known number of data points $n$ from a known distribution $\mathcal{D}$, an additional $kn$ training points from a different distribution $\mathcal{D}'$. |
| Check whether a model $W^*$ was trained on greater than a certain number of data points, in case policy oversight targets the total training compute of a model (e.g. as part of compute usage reporting). | Underreport total training time to avoid triggering oversight. | A test such that, given a target weight checkpoint $W^*$ and claimed sequence of $n$ data points $\Pi$, detects whether the model was in fact trained on $> kn$ data points, for some $k > 1$. |
| Check whether a model $W$ was initialized without using weights obtained from previously-trained models. | A Prover might wish to start training using weights obtained from a previous training run, hiding the fact that more data or compute was used than reported, in order to avoid scrutiny, or to save compute by copying another's work. | A test such that, given a desired initialization $W_0$ (up to hidden unit permutations), makes it cryptographically hard to construct a transcript that results in $W_0$ being an initialization compatible with the resulting transcript. |
| Check whether a model has a backdoor, i.e. an improbable input that yields a disallowed behavior. | An attacker might wish to hide capabilities, or give themselves unauthorized access to systems that will be gatekept by deployed versions of their models. | A test such that, given a transcript, allows reliable detection of backdoors through code or data audits. |
| Check whether a model was trained using at least a certain quantity of data, e.g., as part of a Proof-of-Learning meant to verify the original owner of a model, or to verify that certain safety-best-practice training was done. | A Prover may wish to save on compute costs by doing less training, or to prevent their model from being trained on required data. | A test such that, given a target weight checkpoint $W^*$ and a claimed sequence of $n$ data points $\Pi$, detects whether the model was in fact trained on $< cn$ data points, for some $c < 1$. |
| Check whether a model was trained using a particular datapoint. | A Prover may wish to train on copyrighted content, or uncurated datasets, or obfuscate which training data were used. | A test such that, given a transcript and a target datapoint $x$, detects whether the model was in fact trained on $x$. |

# E  Hardness of Spoofing a Weight Initialization

To recap, by requiring that the Prover initialize a model's weights at a specific value in high-dimensional space $W_0 \in \mathbb{R}^d$ drawn from a pseudorandom vector generator $G_r$, we seek to disallow a class of spoofing attacks based on the Prover hand-picking an initial weight vector $\hat{W}_0$ that will

after training end up close to $W_f$, for example by picking an initialization that is already close to $W_f$ (Attack 2 in [ZLD$^+$22]).

The simplest setting in which defense is impossible, and the Prover can reliably find a random initialization that will converge to a given $W_f$, is in realizable linear models (models with only a single linear layer). Since their loss function is strongly convex, any initialization will converge to a neighborhood of the same final value $W_f$, making it straightforward to construct tweaked datasets with certified-random initializations that result in approximately the same final model. Another counterexample occurs when datasets have a single degenerate solution: it is possible to construct a 2-layer neural network with training data covering the input space and where all the labels are 0, such that the model always converges to a weight vector of all 0s, independent of initialization. We will focus our discussion on the usual case of multi-layer NNs with non-degenerate solutions, as described below.

Below, we will sketch an informal argument that for some radius $r$, for a fixed training data sequence $\Pi$, the probability that a training run initialized at a pseudorandomly-generated [10] weight vector $W_0 = G_r(s)$ ends in a final weight vector $W_f$ that is within distance $r$ of a particular target vector $A$, is less than some small value $\delta < \tilde{o}(1/poly(d))$, where $d$ is the dimension of the neural network. This means that a Prover would need to sample a super-polynomial (in $d$) number of random seeds to find one that would, via training on $\Pi$, result in a fully-valid training transcript that ends close to the weight vector $W_f$ from a previous training run with a different initialization, and therefore that it is exponentially hard to violate the "uniqueness" property from Section 3 if the Prover uses a certified random initialization.

To understand whether this is the case, we can examine the counterfactual claim: that independent of weight initialization, all NNs tend to converge to a small (polynomial) number of modes in weight space. This is indeed the case with linear regression: regardless of the initialization, given sufficient full-rank data all linear model training runs will converge to a neighborhood of the same loss minimum in weight-space. If this were also true for neural networks, then even a small number of randomly-sampled weight initializations would likely yield at least one weight initialization that, after training, converged to a mode close to the target $A$ (assuming $A$ is close to at least one mode, which is the case when $A$ is the outcome of a previous training run $W^f$). Yet, empirically, many works have found that large NNs converge to many different modes [AHS22, FDRC20].

The many modes of the NN loss landscape can be understood through permutation symmetries [AHS22]. Neural networks are equivariant ("equivariant" means that a function changes symmetrically under a group action) under specific permutations of their matrices' columns and rows. Nearly all neural networks have the following permutation symmetries: given a single hidden layer $M_1\sigma(M_2(x))$ where $M_1 \in \mathbb{R}^{a \times b}, M_2 \in \mathbb{R}^{b \times c}$ and $\sigma : \mathbb{R}^b \to \mathbb{R}^b$ is a nonlinearity, and given any permutation matrix $F \in \mathbb{R}^{b \times b}$ (such that $FZ$ permutes the rows of $Z$), then by simple algebra $M_1 F^T \sigma(F M_2 x) = M_1\sigma(M_2 x)$ for all $x$. This means that for any set of successive NN matrices $M_1, M_2$, there are at least $b!$ possible permutations with identical input output behavior. For a neural network $W$ with $k$ nonlinear layers and hidden dimension of each layer $b$, there could be $k-1$ different permutation matrices $F_1, F_2, \ldots$, and we denote to the operation of permuting the flattened weight vector $W$ using a particular value of these $F$s as $P : \mathbb{R}^d \to \mathbb{R}^d$. Each $P$ is drawn from the overall set of valid permutations for a particular architecture $P \in \mathbb{P}(M)$, and we know that $\|\mathbb{P}(M)\| = \Omega\left(2^{kb\log b}\right)$.

A second important property is that gradient descent is itself equivariant under the described permutations. Let $R$ be the training operator, such that $W_f = R(W_0, \Pi)$ is the result of training initial weights $W_0$ on a data sequence $\Pi$. [11] Then it is true that $\forall P \in \mathbb{P}(M)$,

$$P(W_f) = P\left(R(W_0, \Pi)\right) = R(P(W_0), \Pi) = W_f^p$$

where $W_f^p$ is the result of training on the permuted initialization. This is simply a consequence of the fact that the gradient operator commutes with any constant matrix (including the permutation matrix), and that the training process $R$ is comprised of repeated calls to the gradient operator, additions, and scalar multiplications (both of which also commute with the permutation matrix). [12]

---

[10]Assuming that $s$ is chosen randomly, based on assumptions described in Section 4.3.

[11]We omit the inherent noise and hyperparameters inherent in $R$ for brevity.

[12]It is in principle possible to construct optimizers for which this is not the case, but this should hold for all common gradient-based NN training optimizers.

Now, assume that the initialization function $G_r$ is radially symmetric (as is the case with all common initialization schemes, e.g., those based on Gaussians), and therefore the probability that the initialization will start at $W_0$ and $P(W_0)$ is the same for all $P \in \mathbb{P}$. Then the probability that the post-training final weights reach $W_f$ or $P(W_f)$ is also the same. *If* we knew that $\Pr_{W_0 \sim G_r(s)}(\|W_f - P(W_f)\| > 2r) > 1 - \delta$ for some $r$ and small $\delta$, then this derivation would tell us that there are many different weight-space modes into which training could converge, each of which is far apart from the others. (For convenience, let's refer to the number of such far-apart permuted modes as $k$.)

Again, our goal is to show that a random initialization is unlikely to converge after training to within a neighborhood around some vector $A$. Assume that $B$ is one of these modes, and $\|A - B\| < r$.[13] According to the assumption from the previous paragraph on the distance between post-training modes, for any second mode $C$, we know that $\|C - B\| > 2r$ with high probability. By the triangle inequality, we know that:

$$\|A - C\| \geq \|C - B\| - \|A - B\|$$
$$> 2r - r = r$$

Therefore there is some minimum distance $\|A - C\| > r$ between the target $A$ and all other $k$ disjoint modes (each associated with a permutation) of the post-training weight distribution. If the number of such far-apart permutations $k$ is superpolynomial, then no polynomial number of weight initialization samples will result in a final model close to $A$.

However, this argument is predicated on a sometimes-invalid assumption: that there are superpolynomially-many permutations $k = \omega(poly(d))$ of $W_f$, each at least a distance $2r$ from each other. In the case of the counterexample from the beginning, where all initializations converge after training to the weight vector of all 0s, all such permutations are in fact equal, and therefore there is no such distance $r$. Instead, one may need to make an assumption about the non-degeneracy of the distribution of final weight vectors $W_f$, such that permutations of these weight vectors are far apart from each other. We leave analysis of which assumptions fulfill this property as future work. Note that for any specific training transcript which includes a specific $W_f$, the distribution of distances of permutations of $W_f$ can be estimated empirically by manually permuting $W_f$'s matrices.

# F    Experiment Details

For the GPT-2 Experiments we use a cosine learning rate schedule that decays by a factor of 10x by the end of training, with a linear warmup of 2000 steps to a peak learning rate of 0.0006. For the Pythia evaluation experiments, we choose checkpoints from 3 contiguous blocks out of 144 checkpoints: early (first 19 checkpoints), mid (checkpoints at step 62000 to 80000), and late (last 19 checkpoints).

# G    More memorization plots

In the following subsections, we plot memorization $\mathcal{M}$, fraction of points with $\Delta_\mathcal{M}$ above the median, and fraction of points with $\Delta_\mathcal{M}$ below the 10th percentile. For GPT-2, we use 100% of the data to generate Figures 1, 2, and 3, while for Pythia, we use 10% of the data to generate Figure 2. In this section, we show results for smaller sampling rates to highlight that with 1%, or sometimes even 0.1% of the original data, we can still observe the memorization effect.

From Pythia 70M results (Subsections G.1, G.2, and G.3) we can see that as training progresses, the memorization effect becomes less pronounced, such that with a smaller data sampling rate, less of the diagonal get highlighted (Figures 11, 18, 25), and the histograms are closely overlapping (Figure 32) for the last 18 checkpoints. At the same time, we observe that as the model size increases the memorization effect becomes clearer, even with 0.1% data sampling rate. In fact, for the 1B-parameter Pythia model, the memorization effect is still clear for the last few checkpoints (Figures 14, 21, 28, and 35) unlike the 70M-parameter case.

---

[13]If this is untrue for all modes $B$, then by definition there is no initialization that leads close to $A$, which satisfies our original objective of bounding the probability of the final weights converging to a neighborhood of $A$.

## G.1 Memorization

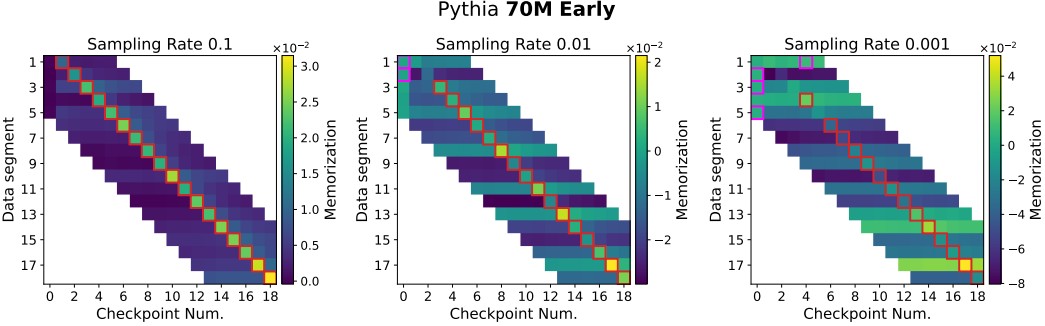

Figure 8: Memorization plots for GPT-2 with different sampling rates.

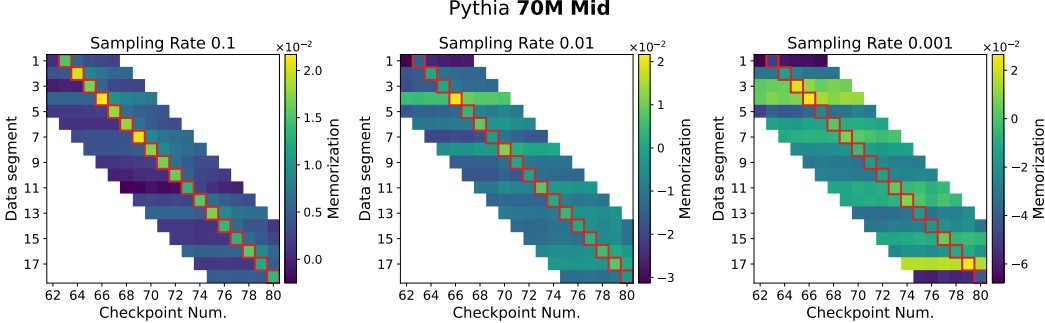

Figure 9: Memorization plots for the first 18 checkpoints of Pythia (70M) with different sampling rates.

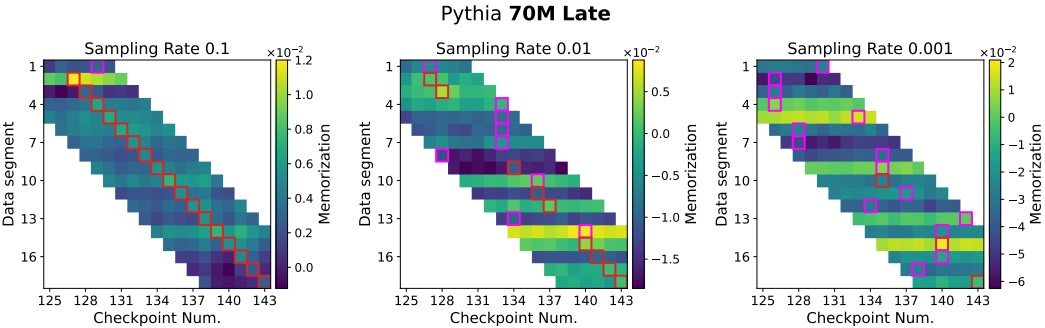

Figure 10: Memorization plots for checkpoints near the middle of Pythia (70M) with different sampling rates.

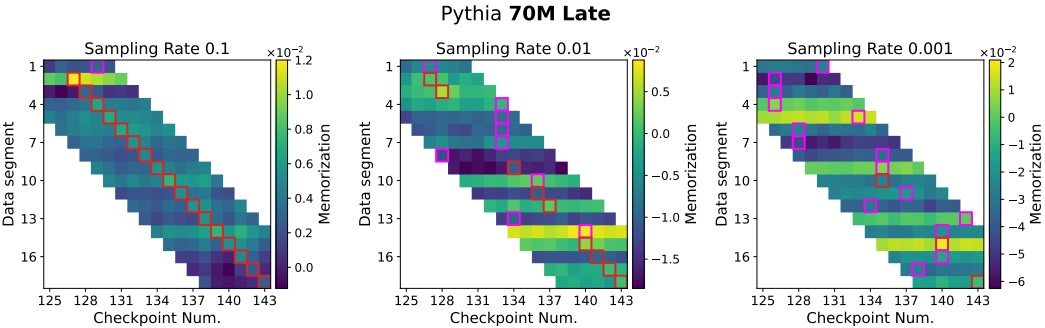

Figure 11: Memorization plots for the last 18 checkpoints of Pythia (70M) with different sampling rates.

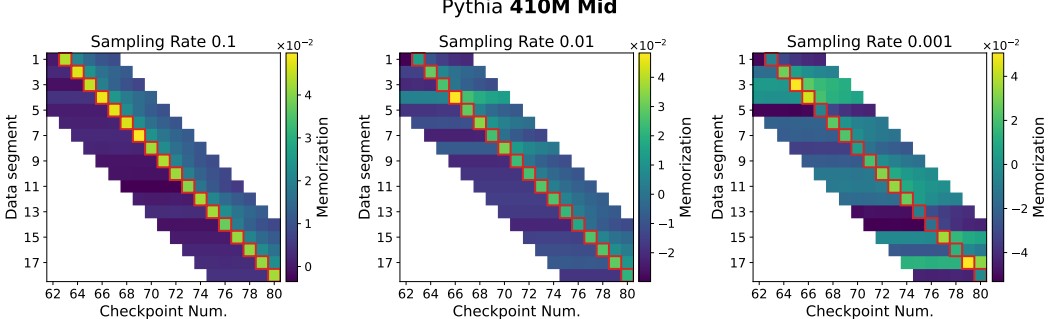

Figure 12: Memorization plots for checkpoints near the middle of Pythia (410M) with different sampling rates.

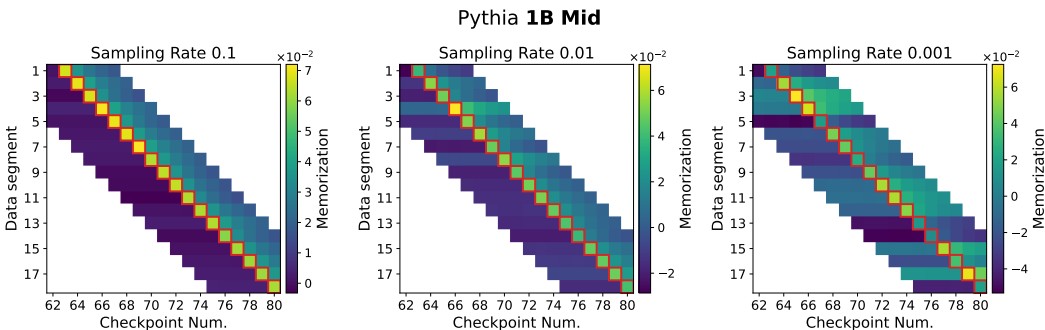

Figure 13: Memorization plots for checkpoints near the middle of Pythia (1B) with different sampling rates.

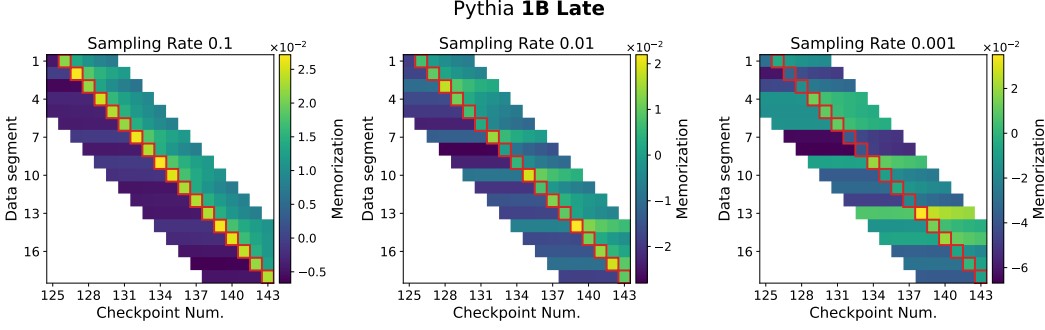

Figure 14: Memorization plots for checkpoints near the end of Pythia (1B) training with different sampling rates.

## G.2 Fraction of Samples Above 50th Percentile

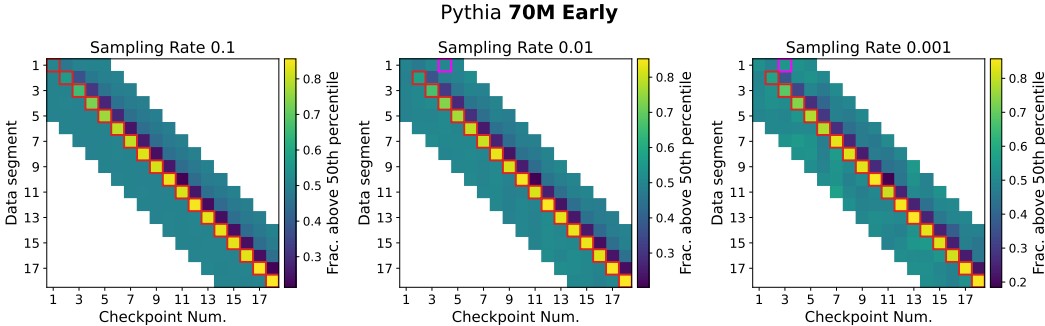

Figure 15: Fraction of samples above the 50th percentile for GPT-2 with different sampling rates.

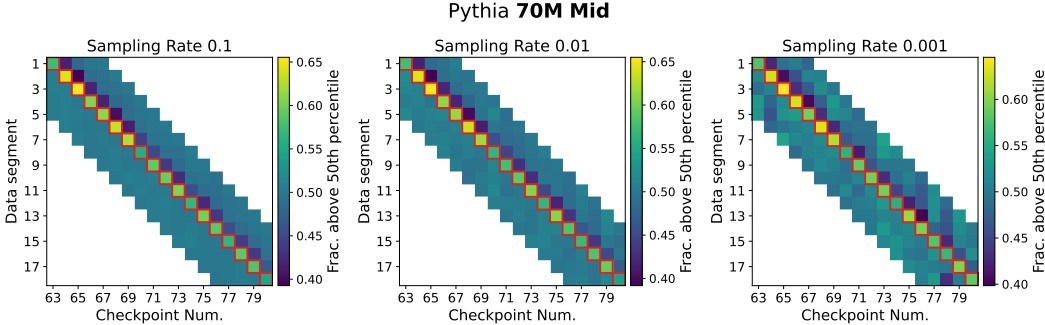

Figure 16: Fraction of samples above the 50th percentile for the first 18 checkpoints of Pythia (70M) with different sampling rates.

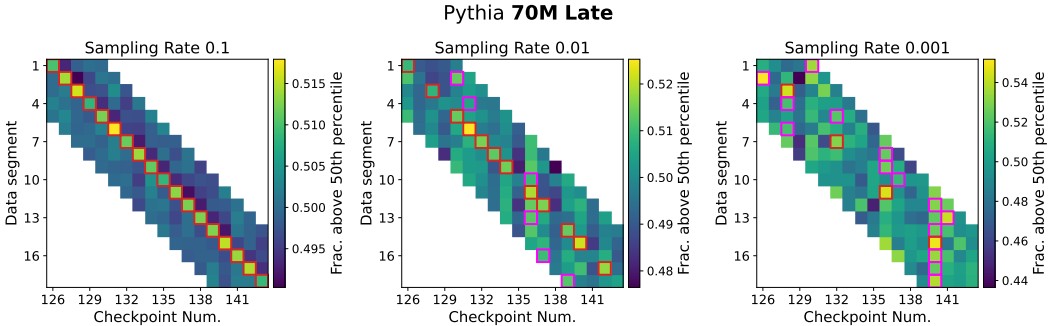

Figure 17: Fraction of samples above the 50th percentile for Pythia (70M) with different sampling rates.

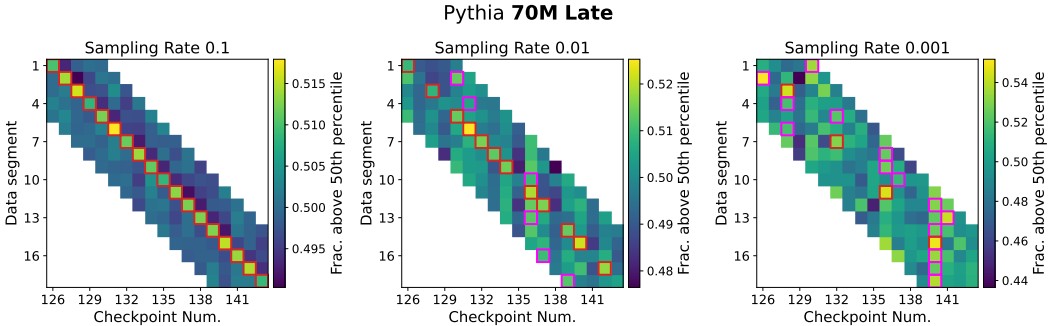

Figure 18: Fraction of samples above the 50th percentile for the last 18 checkpoints of Pythia (70M) with different sampling rates.

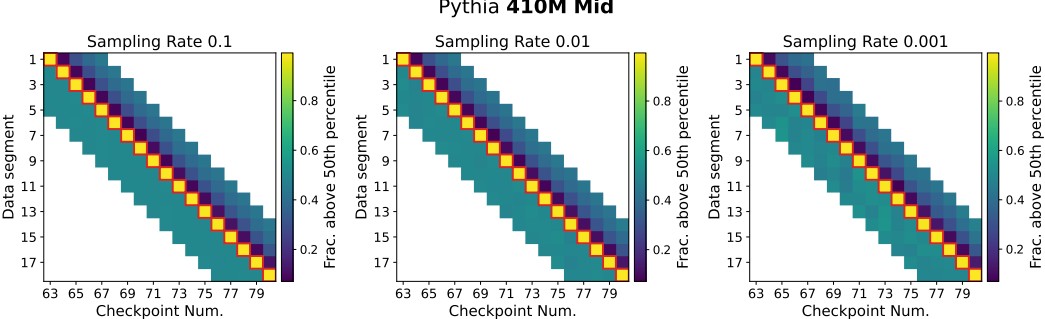

Figure 19: Fraction of samples above the 50th percentile for Pythia (410M) with different sampling rates.

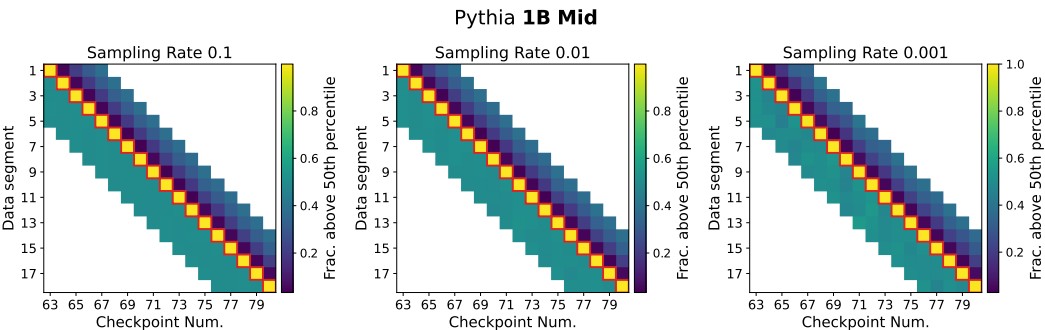

Figure 20: Fraction of samples above the 50th percentile for Pythia (1B) with different sampling rates.

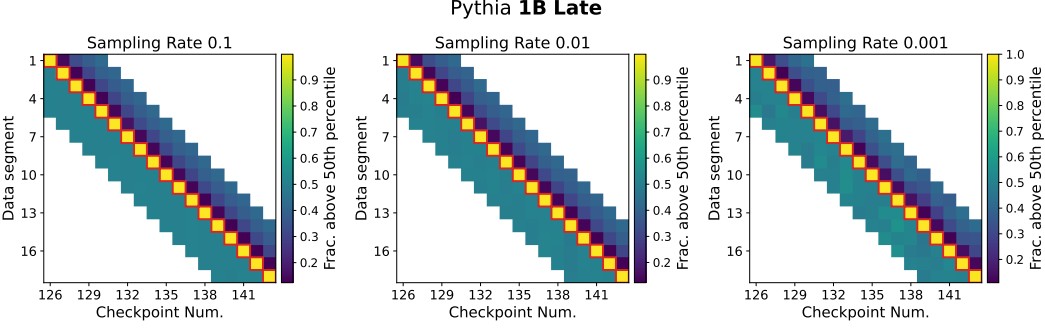

Figure 21: Fraction of samples above the 50th percentile for checkpoints near the end of Pythia (1B) training with different sampling rates.

## G.3    Fraction of Samples Below 10th Percentile

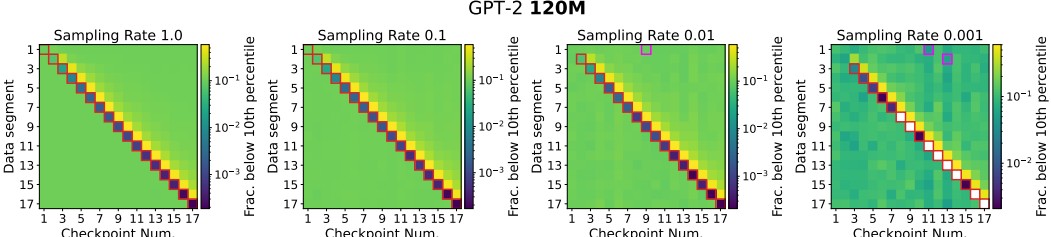

Figure 22: Fraction of samples below 10th percentile for GPT-2 with different sampling rates. White boxes occur whenever the number of samples falling below the 10th percentile is 0.

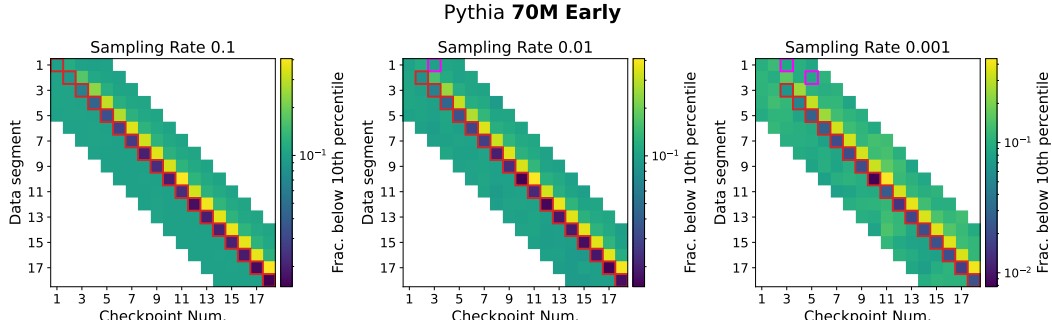

Figure 23: Fraction of samples below 10th percentile for the first 18 checkpoints of Pythia (70M) with different sampling rates. White boxes occur whenever the number of samples falling below the 10th percentile is 0.

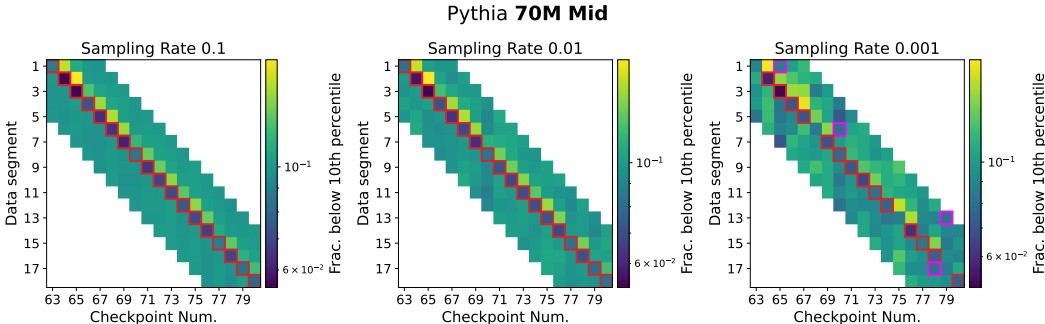

Figure 24: Fraction of samples below 10th percentile for Pythia (70M) with different sampling rates. White boxes occur whenever the number of samples falling below the 10th percentile is 0.

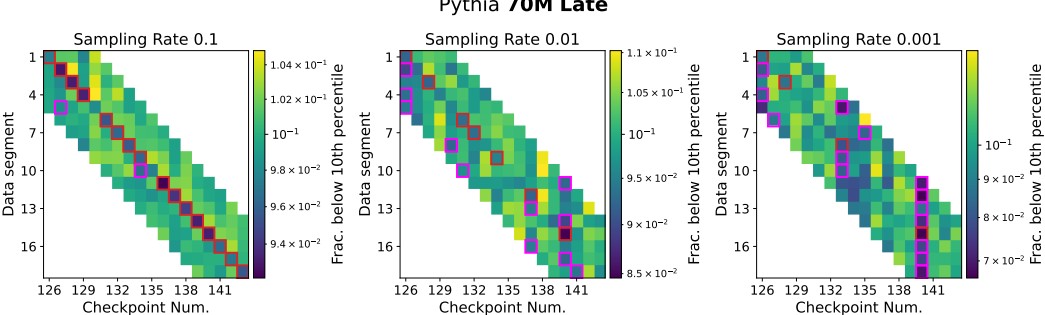

Figure 25: Fraction of samples below 10th percentile for the last 18 checkpoints of Pythia (70M) with different sampling rates.

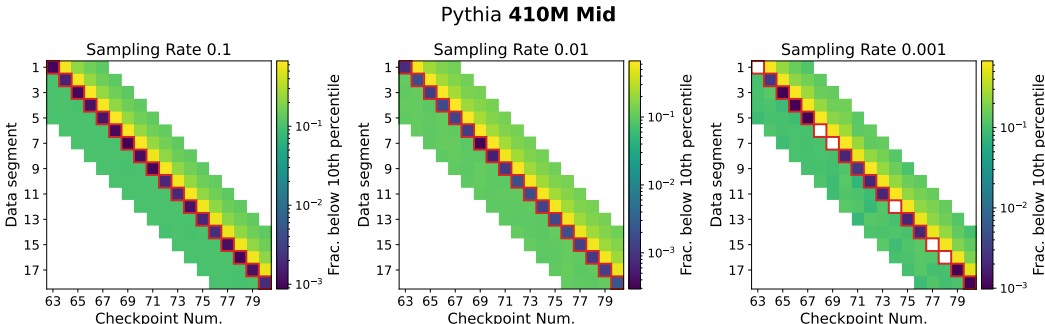

Figure 26: Fraction of samples below 10th percentile for Pythia (410M) with different sampling rates. White boxes occur whenever the number of samples falling below the 10th percentile is 0.

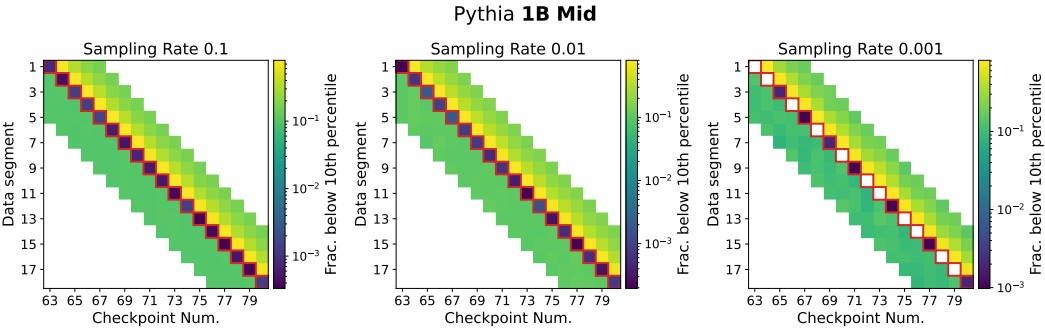

Figure 27: Fraction of samples below 10th percentile for Pythia (1B) with different sampling rates. White boxes occur whenever the number of samples falling below the 10th percentile is 0.

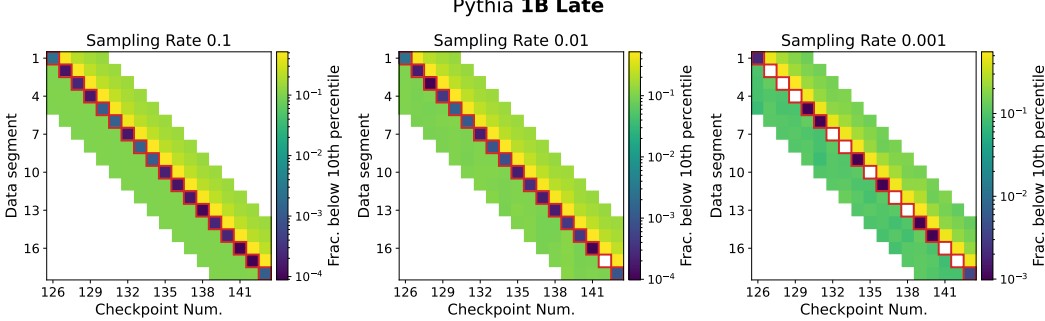

Figure 28: Fraction of samples below 10th percentile for checkpoints near the end of Pythia (1B) training with different sampling rates. White boxes occur whenever the number of samples falling below the 10th percentile is 0.

## G.4 Memorization Delta Histograms

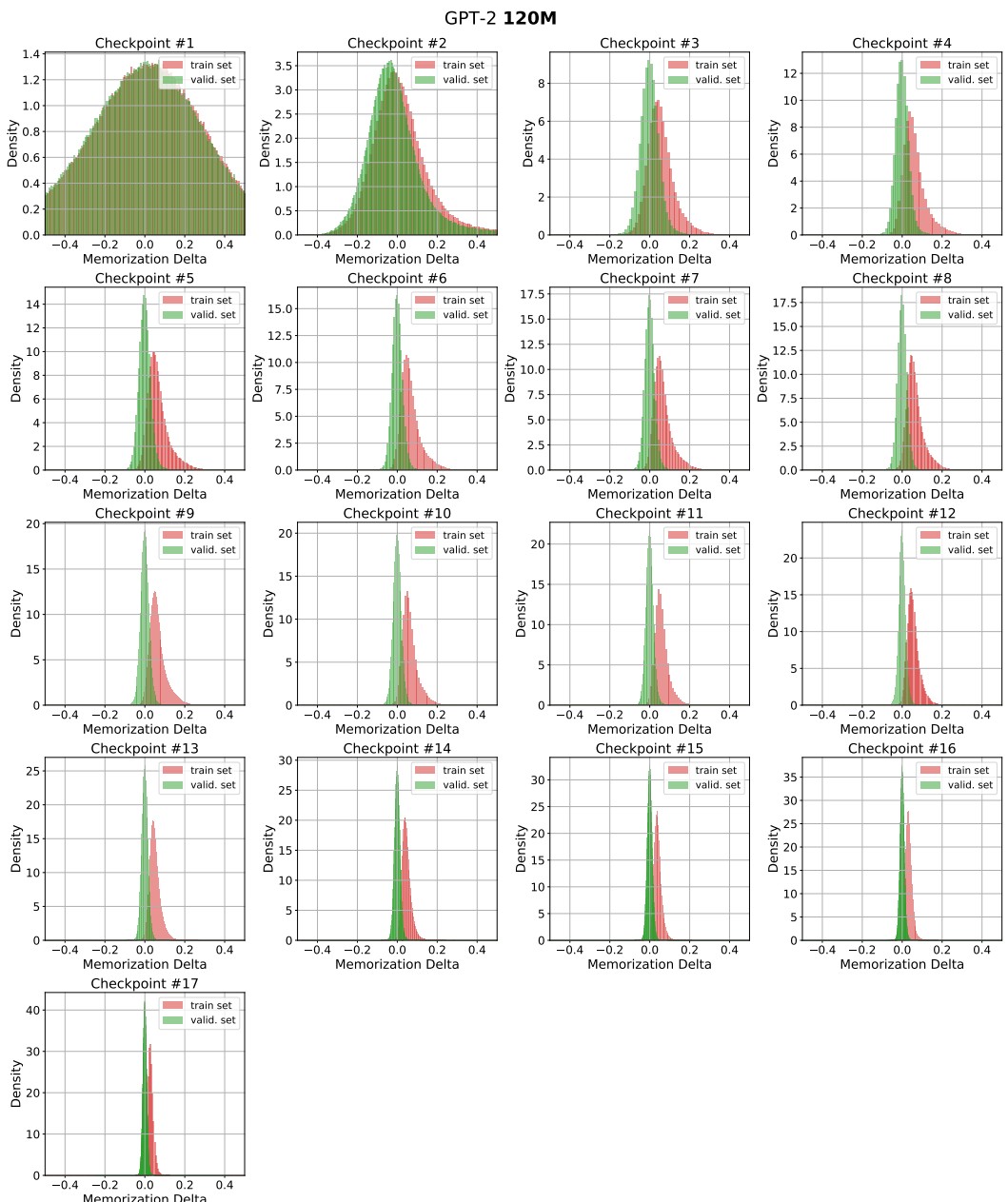

Figure 29: Each subplot compares two histograms of $\Delta_{\mathcal{M}}$, one for $\Delta_{\mathcal{M}}$ resulting from the checkpoint evaluated on its most recent data segment (diagonal), and one for the checkpoint evaluated on the validation set. All checkpoints are from GPT-2 training.

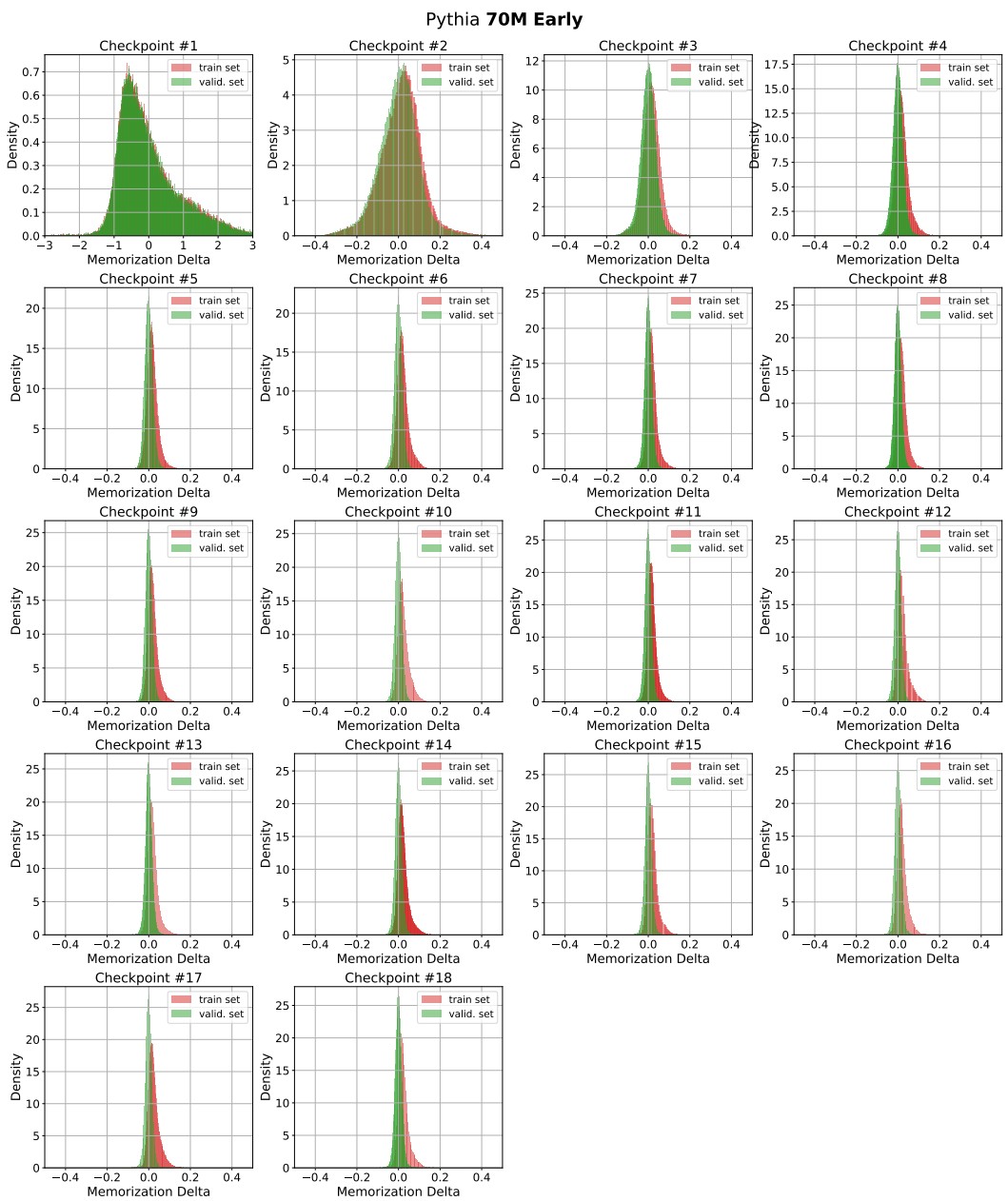

Figure 30: Each subplot compares two histograms of $\Delta_{\mathcal{M}}$, one for $\Delta_{\mathcal{M}}$ resulting from the checkpoint evaluated on its most recent data segment (diagonal), and one for the checkpoint evaluated on the validation set. All checkpoints are the first 18 from Pythia (70M) training.

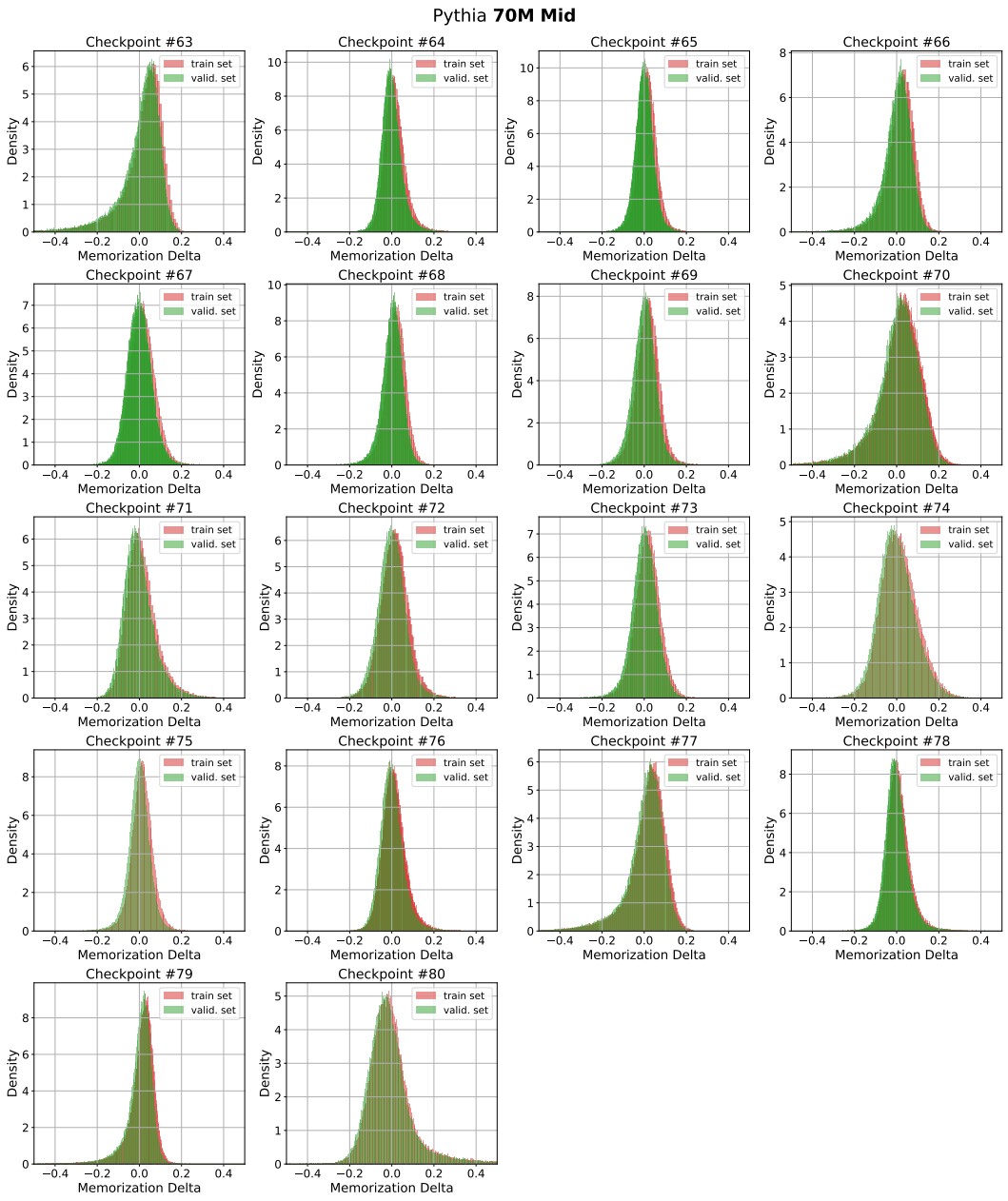

Figure 31: Each subplot compares two histograms of $\Delta_{\mathcal{M}}$, one for $\Delta_{\mathcal{M}}$ resulting from the checkpoint evaluated on its most recent data segment (diagonal), and one for the checkpoint evaluated on the validation set. All checkpoints are from near the middle of Pythia (70M) training.

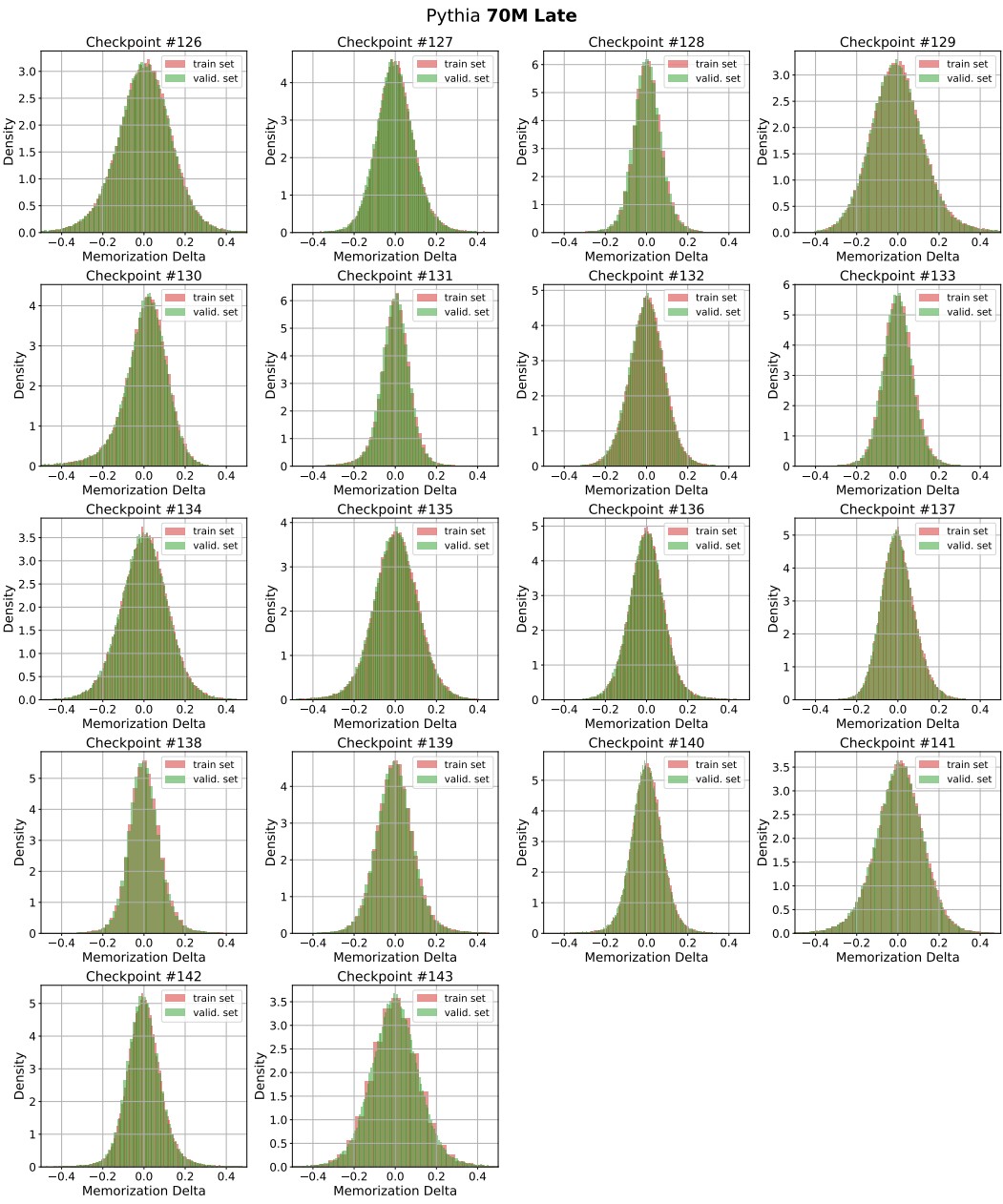

Figure 32: Each subplot compares two histograms of $\Delta_{\mathcal{M}}$, one for $\Delta_{\mathcal{M}}$ resulting from the checkpoint evaluated on its most recent data segment (diagonal), and one for the checkpoint evaluated on the validation set. All checkpoints are the last 18 from Pythia (70M) training.

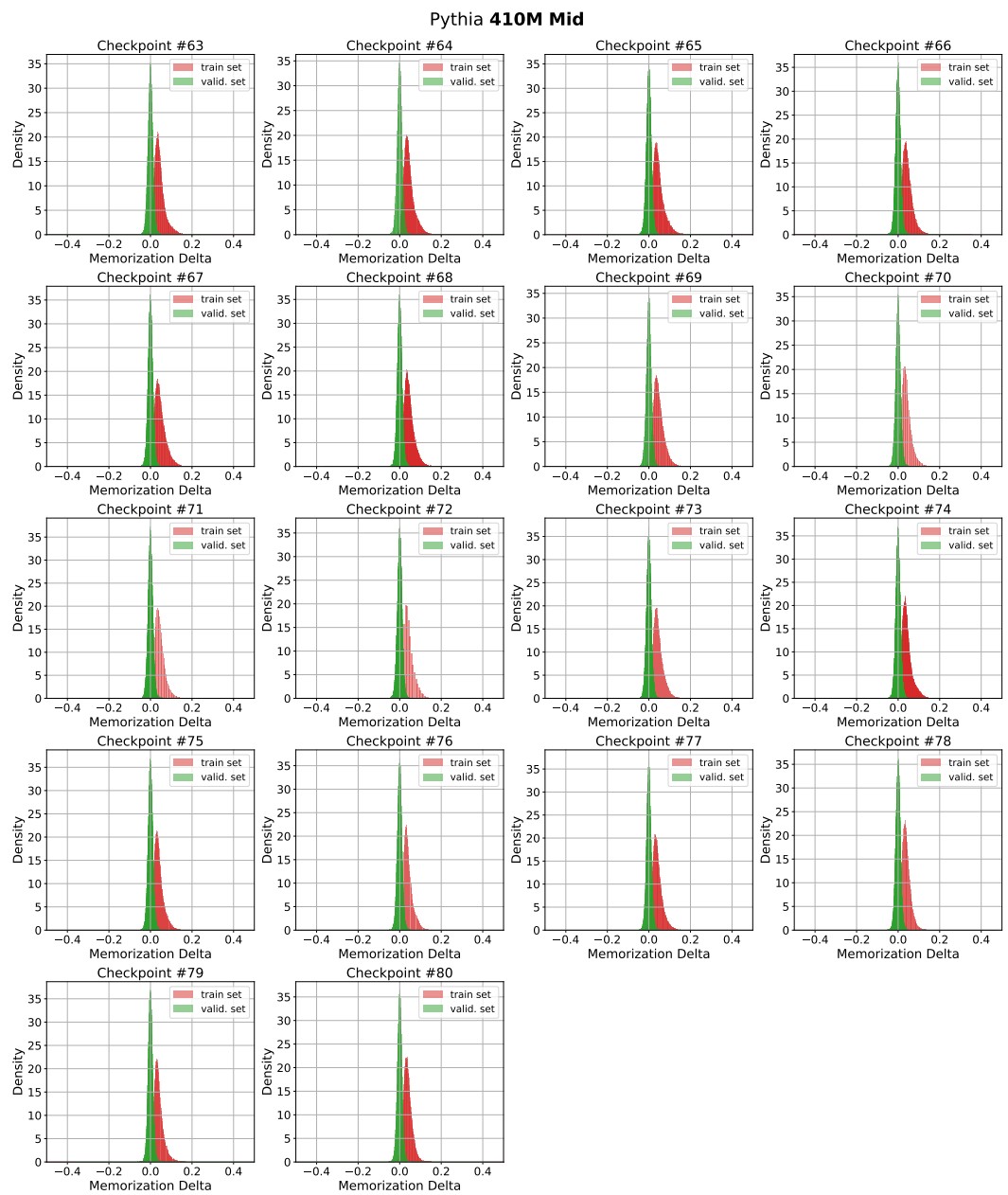

Figure 33: Each subplot compares two histograms of $\Delta_{\mathcal{M}}$, one for $\Delta_{\mathcal{M}}$ resulting from the checkpoint evaluated on its most recent data segment (diagonal), and one for the checkpoint evaluated on the validation set. All checkpoints are from near the middle of Pythia (410M) training.

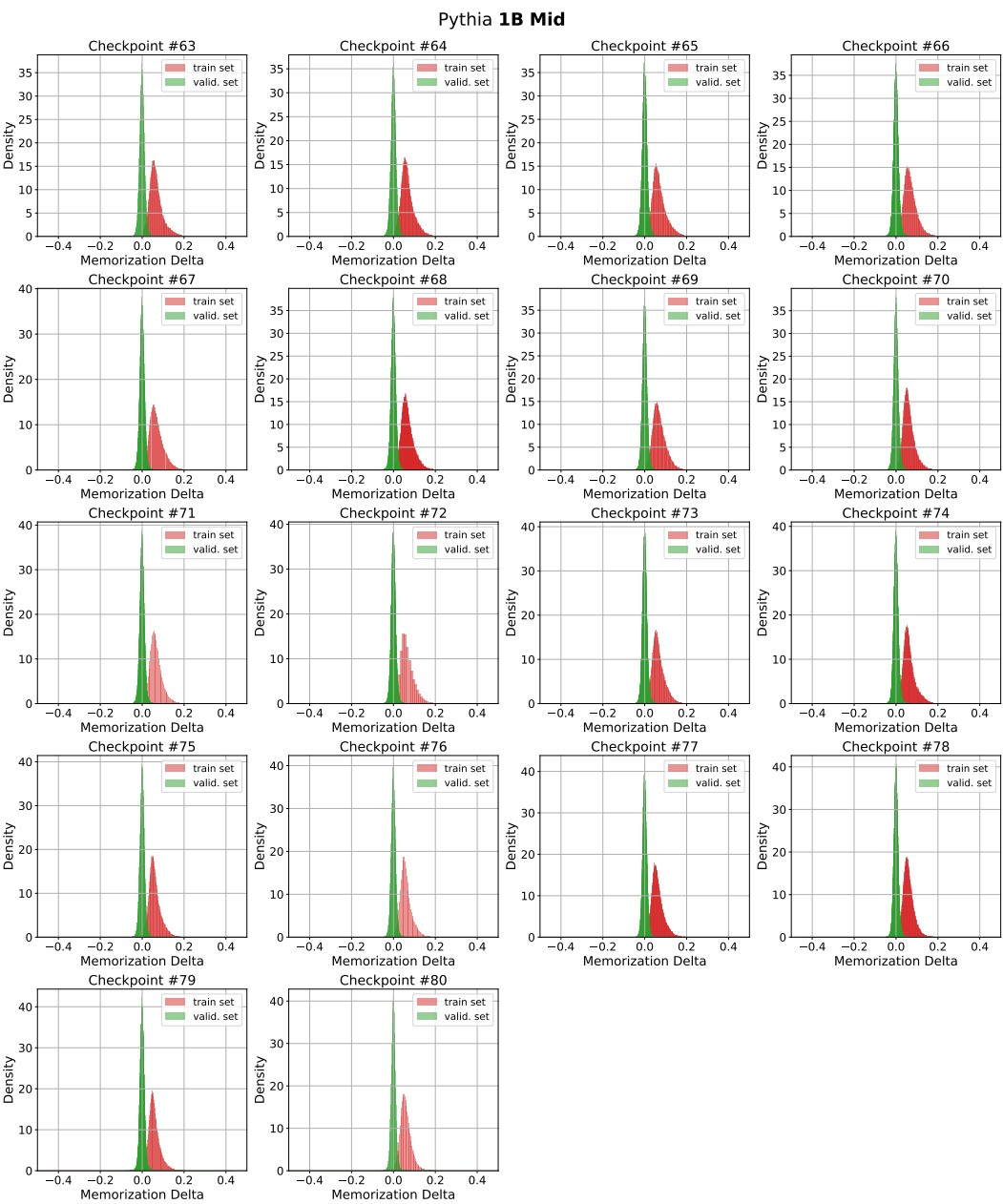

Figure 34: Each subplot compares two histograms of $\Delta_{\mathcal{M}}$, one for $\Delta_{\mathcal{M}}$ resulting from the checkpoint evaluated on its most recent data segment (diagonal), and one for the checkpoint evaluated on the validation set. All checkpoints are from near the middle of Pythia (1B) training.

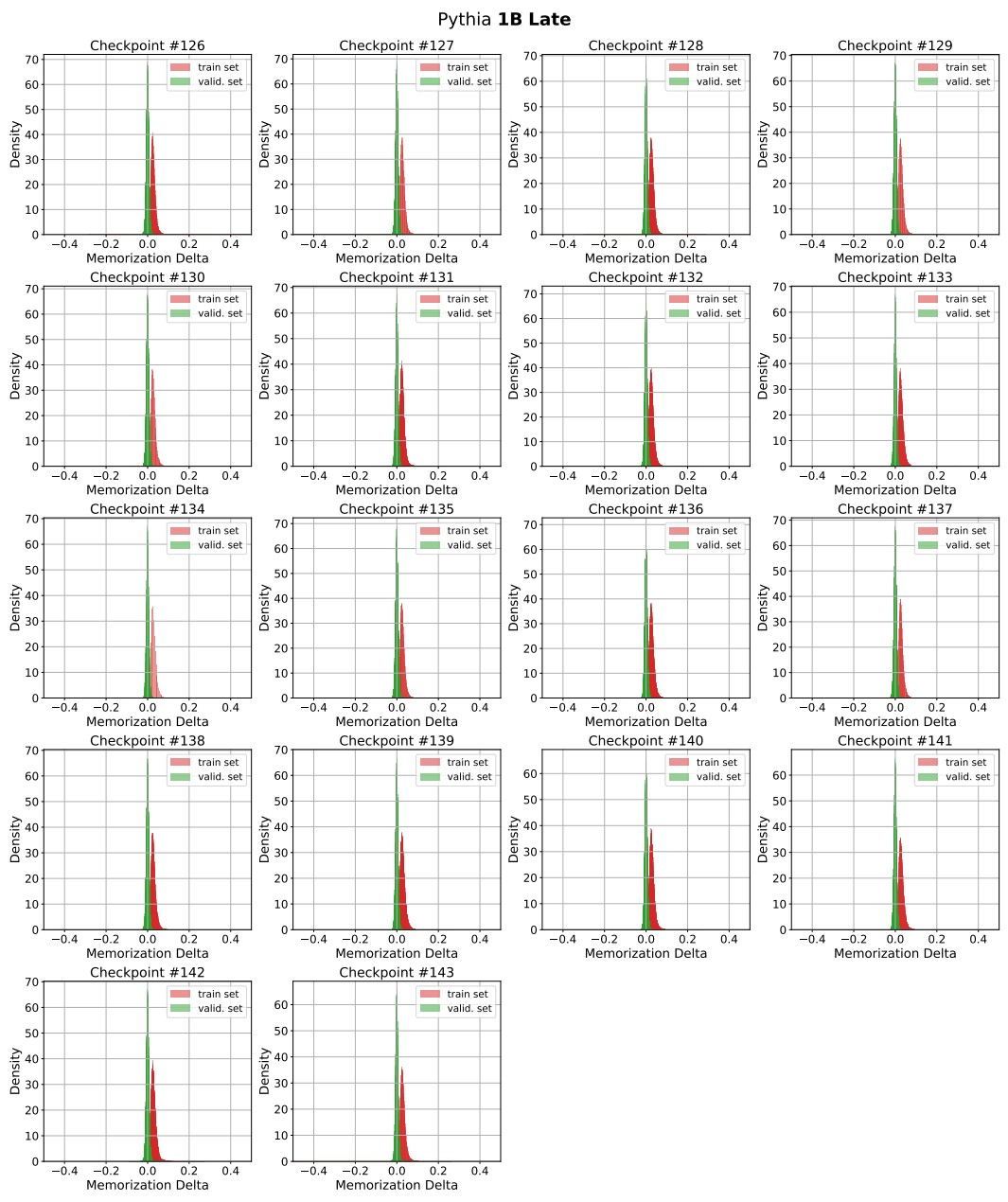

Figure 35: Each subplot compares two histograms of $\Delta_{\mathcal{M}}$, one for $\Delta_{\mathcal{M}}$ resulting from the checkpoint evaluated on its most recent data segment (diagonal), and one for the checkpoint evaluated on the validation set. All checkpoints are from near the end of Pythia (1B) training.

# H More attack plots

## H.1 Interpolation Attack

We repeat the interpolation attack experiment in the main body of the paper, with the 1B-parameter Pythia model and observe from Figure 36 that indeed the interpolated checkpoints fail our memorization tests.

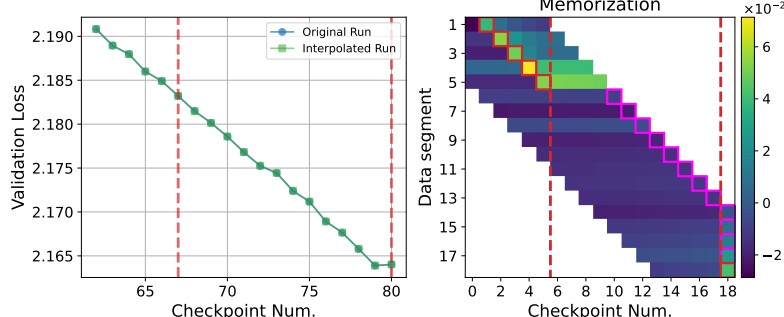

Figure 36: Simulating an interpolation attack by training a Pythia (1B) model until the 67th checkpoint, and then linearly-interpolating to the 80th checkpoint. On the left, we show that an attacker can carefully choose interpolation points to mask any irregularities in validation loss. (The green line perfectly overlaps with the blue line.) Nonetheless, on the right, we see a clear signature in the memorization plot, computed using only 1% of data: the typical memorization pattern along the diagonal does not exist for the interpolated checkpoints. For each row corresponding to a data segment $\Pi_i$, a box marks the maximal-$\mathcal{M}$ checkpoint. The box is red if the checkpoint is a match $W_i$, and magenta if there is no match and the test fails $W_{j \neq i}$.

## H.2 Data Subtraction Attack Tests

In the following subsections, we plot the subtraction-upper-bound heuristic $\lambda(\Pi_i, p, W_i)$ with varying values of $p$, for different subtraction rates. We observe that for big enough models $\lambda$ is a tight upper-bound when no subtraction has happened. For Pythia with 70M parameters, our smallest model, $\lambda$ does not provide a tight upper-bound. However, for GPT-2 with 124M parameters, Pythia with 410M parameters, and Pythia with 1B parameters, $\lambda$ provides a tight upper-bound.

For the 1B-parameter Pythia model, we further plot the upper-bound heuristic for varying values of the checkpoint interval (number of training steps between each checkpoint). From Figure 41, we observe that even though $\lambda$ increases as the interval increases, it is still a good upper-bound ($\sim 0.05$ for a checkpoint interval of 5000 steps) for $p = 0.1$ and $p = 0.2$. This means that we can save checkpoints less frequently and still use the heuristic to detect data subtraction.

### H.2.1 GPT-2

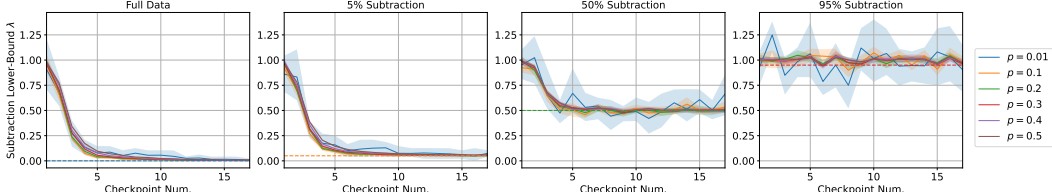

Figure 37: The subtraction-upper-bound heuristic $\lambda(\Pi_i, p, W_i)$ is robust to different values of $p$, especially for the full data case. The heuristic was computed using just 1% of training data, across 20 random seeds, with dashed lines showing the true subtraction rate.

### H.2.2 Pythia (70M)

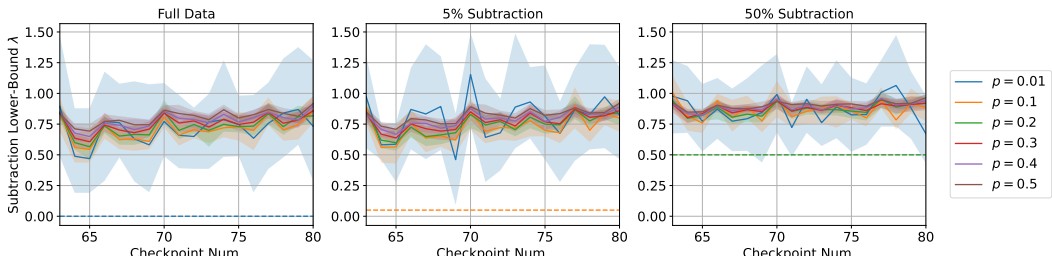

Figure 38: The subtraction-upper-bound heuristic $\lambda(\Pi_i, p, W_i)$ does not provide a good upper bound for a smaller model (Pythia with 70M parameters). The heuristic was computed using $1\%$ of training data, across 20 random seeds, with dashed lines showing the true subtraction rate.

### H.2.3 Pythia (410M)

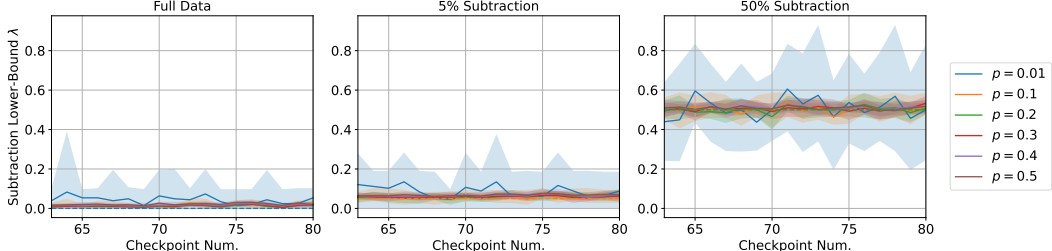

Figure 39: The subtraction-upper-bound heuristic $\lambda(\Pi_i, p, W_i)$ is a good upper bound for a big enough model (Pythia with 410M parameters). The heuristic was computed using $1\%$ of training data, across 20 random seeds, with dashed lines showing the true subtraction rate.

### H.2.4 Pythia (1B)

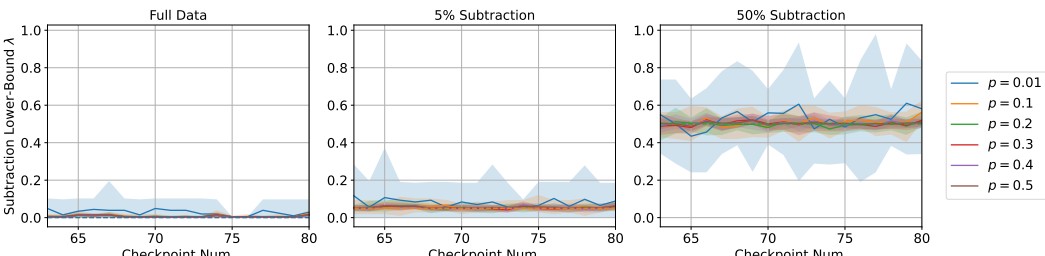

Figure 40: The subtraction-upper-bound heuristic $\lambda(\Pi_i, p, W_i)$ is a good upper bound for a bigger model (Pythia with 1B parameters). The heuristic was computed using $1\%$ of training data, across 20 random seeds, with dashed lines showing the true subtraction rate.

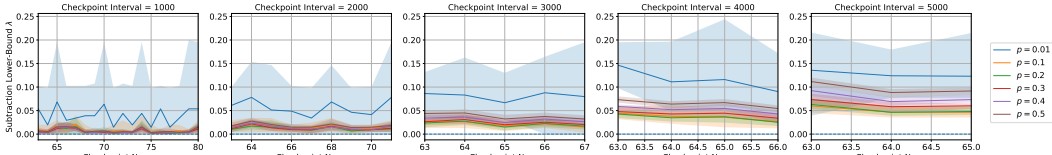

Figure 41: Even though the subtraction-upper-bound heuristic $\lambda(\Pi_i, p, W_i)$ increases as number of steps between checkpoints increases, it still provides a good upper bound for $p = 0.1$ and $p = 0.2$. The heuristic was computed using $1\%$ of training data, across 20 random seeds, with dashed lines showing the true subtraction rate.

# I   Broader Impacts

We intend this work to be a step towards meaningful and transparent public oversight of large AI systems, especially those with capabilities whose irresponsible use could significantly harm the public. Our protocol is a sketch of a technical framework for a system by which AI developers can prove properties of their training data, and may thereby enable the effective enforcement of a broader set of policies than those solely relying on querying models "black-box". While enabling many possible positive rules, this could also be misused by coercive states to detect and enforce harmful restrictions on beneficial AI development. However, in most cases, such authoritarian states would already have a means for policing domestic AI developers' behavior, and verification tools demanding so much cooperation from the Prover are unlikely to meaningfully increase existing surveillance powers. Another issue is that requirements for complying with monitoring and enforcement tend to favor large companies, for whom the cost of compliance can more easily be amortized. This motivates efforts to keep verification schemes simple, flexible and cheap.

We hope that this protocol can also be useful for verifying agreements between untrusting countries. The protocol itself does not provide a means for identifying that an AI model was developed in the first place unless it is disclosed. In this sense, it more closely parallels a process for an AI-developing country to allow its counterpart to retroactively inspecting a developed system (paralleling the New START treaties' inspections of nuclear launchers), rather than to proactively detect when a new system is deveoped (paralleling the IAEA's monitoring of the process of uranium enrichment).

Because our protocol supports multiple independent auditors reviewing the same transcripts, we hope that these tools will support the development of trust between competing companies and countries. Ultimately we hope such protocols will support the development of a larger governance ecosystem representing many parties.

