# OpenReview forum: "Tools for Verifying Neural Models' Training Data"
_NeurIPS.cc/2023/Conference — NeurIPS 2023 poster_

### Official Review · Reviewer_pydU · 2023-06-21

**Soundness:** 2 fair
**Presentation:** 2 fair
**Contribution:** 3 good
**Rating:** 5
**Confidence:** 2

**Summary:**

This paper proposes a protocol for verifying that a model trainer submits data and learned weights, and a verifier checks whether the weights are correctly learned from the submitted data.
Such a protocol is useful for trustworthy AI. The paper defines the problem of Proof-of-Trainind-Data (PoTD). It is inspired by the previous work on "Proof-of-Learning," but the setting of PoTD requires more. The paper gives a formal definition of PoTD protocol and argues that the protocol needs to satisfy some conditions to achieve the guarantee in a practical setting.

**Strengths:**


The motivation of the paper is interesting. The proposed PoTD protocol seems reasonable.

**Weaknesses:**


1. The paper does not explain the motivation why we need the PoTD protocol and its impact. The paper says that some attacks can be treated by existing Proof-of-Learning methods. It seems trivial since  PoTD poses stronger requirements, as written in the paper. The paper should discuss the problems of existing PoL methods and how the proposed protocol can solve them.

2. The only attack that the existing method cannot deal with but the proposed method can is a data subtraction attack. How important to deal with such an attack is not explained enough.
Moreover, experimental results show how the proposed method performs but do not compare with baseline methods. Therefore, experiments are not enough to
show the effectiveness of the proposed methods.

3. The presentation of the paper is hard to follow. The paper seems to consist of fragments of texts relationships between them are unclear.
For example, There is a formal definition of PoTD protocol in section 2, but the definition is never mentioned in the subsequent sections. Therefore, it is hard to judge whether the definition is reasonable or not.
The memorization heuristic introduced in Section 3.2 seems overly complex. If we use $-L(d, W)$ instead of M(d, W) in (3), then the values of PBQ and FBQ would not change since
the first term of (2) does not depend on data d and has no effect when evaluating $\Delta_M(d^\prime, W) >  \Delta_M(d, W)$.



**Questions:**

n/a

**Limitations:**

The paper mentions to limitations in the end of the paper.

---

> ### Author Rebuttal · Authors · 2023-08-10
>
> Thank you for your feedback. Below, we respond specific points made in your review:
>
> **W1:** *The paper says that some attacks can be treated by existing Proof-of-Learning methods. It seems trivial since PoTD poses stronger requirements, as written in the paper. The paper should discuss the problems of existing PoL methods and how the proposed protocol can solve them.*
>
> **Response:** We are not sure which statements you are referring to. Most of the attacks raised in the paper cannot be affordably addressed using existing Proof-of-Learning methods, and even for those that can, our solutions are more efficient or have other benefits (e.g. only requiring inference and not retraining). We provide specific examples of attacks that PoL methods cannot address, but that our new methods can:
> - **PoL-style retraining is known to be potentially vulnerable to synthetic data attacks** [1], which allow the Prover to swap the real data out for carefully-crafted fake data. This is fatal for PoTD, and can also enable PoL spoofs (see [1]). Our new verified-data-shuffling defense in Section 3.3 stops this attack.
> - **Retraining is inefficient for catching small data lies**. Given 100 checkpoints, if a Prover wants to misreport 1 checkpoint’s worth of data, then to catch them w/ 80% probability the Verifier would have to rerun 80% of the original training run. However, with our method, the Verifier needs to spend as little as 1% (and only inference compute) to catch subtraction/interpolation attacks, and slightly more for addition attacks (which require retraining a couple segments to calibrate segment-magnitudes, but still yield a roughly order-of-magnitude improvement). Even if a Verifier is only worried about Provers misreporting 10 checkpoints, the Verifier would need to retrain $\approx 16\%$ of the checkpoints (see Appendix C of [2] for a similar derivation) to catch the lying Prover with 80% probability, much larger than the 1% or few percent required by our method.
>
> We will include the above examples to better highlight the shortcomings of prior methods at achieving PoTD.
>
> [1] Zhang, Rui, et al. "“Adversarial Examples” for Proof-of-Learning." 2022 IEEE Symposium on Security and Privacy (SP). IEEE, 2022.
>
> [2] Shavit, Yonadav. "What does it take to catch a Chinchilla? Verifying Rules on Large-Scale Neural Network Training via Compute Monitoring." arXiv preprint arXiv:2303.11341 (2023).
>
> **W2a:** *The only attack that the existing method cannot deal with but the proposed method can is a data subtraction attack. How important to deal with such an attack is not explained enough.*
>
> **Response:** To clarify, our methods not only enable detection of data subtraction attacks, but also block synthetic-data-substitution attacks (see Section 3.3, and [3] for the paper that originally provided the attack), and are the first to enable cost-effective detection of interpolation attacks and large-scale data addition attacks. Together, these block an attacker from far-underclaiming the training data they used, which is directly relevant for, e.g., policies around the amount of compute used for training. We will make this clearer in the paper.
>
> [3] Zhang, Rui, et al. "“Adversarial Examples” for Proof-of-Learning." 2022 IEEE Symposium on Security and Privacy (SP). IEEE, 2022.
>
> **W2b:** *Moreover, experimental results show how the proposed method performs but do not compare with baseline methods.*
>
> **Response:** Most of our defenses have no relevant baselines, such as data subtraction attacks on individual segments. In principle, one could run PoL-style retraining at a massive scale, but this would be far too expensive to be practically relevant.
>
> **W3a:** *There is a formal definition of PoTD protocol in section 2, but the definition is never mentioned in the subsequent sections.*
>
> **Response:** To clarify the paper’s structure, Section 2 provides a theoretical definition for PoTD, but proving such a theoretical guarantee is in practice currently impossible for large-scale NN training. Instead, just as in the original PoL paper, in Section 3.3 and 5 we propose a set of attacks, which are intended to approximate the full set of attacks on PoTD. We then show that our solution is robust to these attacks, and thereby demonstrate that our defenses heuristically satisfy Definition 1, for current attacks in the literature. We will clarify the set of considered attacks explicitly in Section 2, and enumerate all the attacks in a single location.
>
> **W3b:** *The memorization heuristic introduced in Section 3.2 seems overly complex. If we use $-L(d, W)$ instead of M(d, W) in (3), then the values of PBQ and FBQ would not change since the first term of (2) does not depend on data d and has no effect when evaluating $\Delta_\mathcal{M}(d’,W) > \Delta_\mathcal{M}(d, W)$.*
>
> **Response:** You are correct that the existence of the normalization term $E_{d' \in D_v}[L(d', W)]$ in our memorization heuristic $\mathcal{M}$ does not affect the values of PBQ and FBQ. This could represent a 2x efficiency gain, and we will mention it in the final manuscript. This normalization may still be important in other cases for preserving the meaning of the Memorization Delta $\Delta_\mathcal{M}$, which is used to produce memorization-charts. Without the normalization term, it becomes hard to compare $\Delta_\mathcal{M}$ across segments, especially in the earlier stages of training when the changes in loss between checkpoints vary quite a bit across segments. To illustrate, we created Memorization Delta plots with and without the normalization term to illustrate this point (see attached PDF).

---

> > ### Comment · Reviewer_pydU · 2023-08-14
> > **Thank you for the response**
> >
> > Thank you for the response. The claims in the response are all strong, and I understand that the proposed method has many interesting properties compared with PoL methods.
> > My question is, "Why did the authors not put these things in the original paper?"
> > I think the paper's introduction does not clearly show the motivation of the PoTD. It says PoTD is "inspired by PoL (line 33)", and "provide several verification strategies ... all
> > published attacks in the PoL literature (line 37)". Moreover, the paper also says PoTd is a "stricter requirement than POL" (line 52).
> > However,  the introduction says nothing about why we need PoTD instead of PoL. These things made it hard for me to understand the position of PoTD.
> >
> > I will raise my score, but I still think this paper needs a major revision to improve the presentation by including the material shown in the response.

---

### Official Review · Reviewer_G7nA · 2023-06-26

**Soundness:** 3 good
**Presentation:** 2 fair
**Contribution:** 2 fair
**Rating:** 5
**Confidence:** 2

**Summary:**

The authors propose Proof of Training Data (PoTD), a variant of Proof-of-Learning (PoL) protocols that focuses on training set attacks, rather than the training algorithm itself. A valid PoTD protocol should be able to, at least in theory, spot when a machine learning model has been trained on a different training set than the one declared by the learner. As in PoL, the learner is required to provide a full transcript of the training process, including training data, code and intermediate checkpoints. Unfortunately, the task of verifying a training transcript is as computationally intensive as re-training the model from scratch. For this reason, the authors propose several heuristic strategies for PoTD, which rely on the fact that stochastic gradient ascent tends to first memorize and then forget the data it observes in each batch.

**Strengths:**

From the methodology standpoint, I truly appreciate the memorization heuristics proposed in the papers. In particular, they seems to be able to defend against a large number of different threat models.

**Weaknesses:**

The PoTD protocol proposed by the authors have considerable overlap with the existing PoL proposals. I am inclined to see it as a variant of these existing efforts, rather than a novel, independent idea.

Furthermore, the defenses proposed in the paper are heuristics and have been tested on large language models only. The authors mention that their techniques may work differently on other neural architectures, learning tasks and training procedures. Therefore, I believe more experimentation is needed to confirm their usefulness.

Additionally, the PoTD protocol requires the learner to disclose their complete learning process. Thus, there is no way to protect the intellectual property of the learner. The authors are honest about this limitation, and claim it will be addressed as future work. However, I feel this makes the paper weaker.

The data addition attack presented in lines 256-267 seems the most interesting scenario to me. It is unfortunate that the heuristic technique proposed by the authors cannot defend against it.

The writing style of the paper could be improved. I report here some specific examples.

Line 6, "and flag if the model specific harmful or beneficial data sources" is not a syntactically-correct English sentence.

Line 30, the authors start discussing how to solve proof-of-training-data
before giving a precise enough definition that the reader can follow.

Line 87, c2 is missing from the definition of V (compare with Line 82). Also, why is the probability taken over c1, when c1 does not appear in any of the terms?

Line 172-173, please number all equations.

Section 3.3 is very dense and many important details are left for future work.

Footnotes 4-9 occupy almost a quarter of the page and contain important information. I would prefer having them merged with the main text.

Line 225, "on trained" should be "trained on".

Section 5 introduces new concepts, new attacks, new defenses, new notation. Since this happens so late in the paper, it ends up being a bit overwhelming. Why not explicitly organise the whole paper by threat model?

**Questions:**

Would it be possible to bypass the memorization heuristic by changing both the model and the data? Theoretical work has shown that only a small portion of a large neural network is important for achieving good predictive performance (lottery tickets). Once a learner has trained a good model on dataset D*, it should be possible to add (a large number of) redundant neurons and retrain those on D. Would this be a valid attack on PoTD?

The Equation between lines 172 and 173 is not obvious at first sight, and should be clarified. Also, please number all equation in the paper.

**Limitations:**

I commend the authors for beign upfront about the limitations of their work. All my concerns have been listed above.

---

> ### Author Rebuttal · Authors · 2023-08-10
>
> Thank you for your feedback. Below, we respond specific points made in your review:
>
> **W1:** *The PoTD protocol proposed by the authors have considerable overlap with the existing PoL proposals. I am inclined to see it as a variant of these existing efforts, rather than a novel, independent idea.*
>
> **Response:** We emphasize that though our solution shares a common structure with techniques from the PoL literature, the PoTD problem is fundamentally different, and requires different tools.
> First, we clarify the difference between PoL and PoTD. **PoL only checks if the Prover is capable of spending the compute** to perform a large training run, but does not check whether the training transcript disclosed **actually corresponds** to the training run that yielded $W^*$. For example, a Prover can do an original training run, and then replace 10% of the original training data with synthetic data as in [1] (e.g. to hide that data from the Verifier), and the resulting modified transcript **would still be a valid PoL**. PoTD is more ambitious in that it **checks whether the exact reported data transcript would actually result in** $W^*$. We will better highlight this distinction in the camera ready.
>
> We also emphasize that existing methods from the literature, while being close to achieving PoL, are far from achieving PoTD, and our new methods are important for closing this gap. For example, segment-wise retraining fails to stop replacement-with-synthetic-data attacks, which are a major problem for PoTD (but less fatal for PoL); we address this with our defense in Section 3.3. Prior methods are also too inefficient to practically catch attacks that target a small fraction of segments (e.g. addition, subtraction, or few-segment interpolation), as a PoL-style Verifier could not afford to retrain a large fraction of original training segments to spot a few spoofed ones.
>
> **W2:** *Furthermore, the defenses proposed in the paper are heuristics and have been tested on large language models only. The authors mention that their techniques may work differently on other neural architectures, learning tasks and training procedures. Therefore, I believe more experimentation is needed to confirm their usefulness.*
>
> **Response:** We emphasize that our experiments on Pythia models provide validation of our method’s performance “in the wild”, as they used different architectures and training procedures (and thus are as close to a “random alternative draw of hyperparameters” as we can practically get). We encourage future work on additional modalities and architectures, and have prioritized large transformers as they are the primary focus of current regulatory discussions.
>
> **W3:** *Additionally, the PoTD protocol requires the learner to disclose their complete learning process. Thus, there is no way to protect the intellectual property of the learner. The authors are honest about this limitation, and claim it will be addressed as future work. However, I feel this makes the paper weaker.*
>
> **Response:** This assumption of Verifier access to training data and model checkpoints at verification-time is the standard assumption used in all prior works in the Proof-of-Learning literature. In fact, in contrast to prior defenses (such as retraining), our memorization-defense and data-order-defense do not require knowledge of the training hyperparameters, and are thus essentially out-of-the-box verifiable using only black-box API access, preserving confidentiality. We will update the Discussion to better clarify this contribution. (Catching data addition attacks still does require weights and training-hyperparameters access, hence our comment on the need for future work.)
>
> **W4:** *The data addition attack presented in lines 256-267 seems the most interesting scenario to me. It is unfortunate that the heuristic technique proposed by the authors cannot defend against it.*
>
> **Response:** We wish to clarify that our method does allow us to catch at-scale data addition attacks, so long as they are a significant fraction of the training data in that segment, e.g. >5%. We strongly concur that better defenses against data addition attacks would be an excellent focus of future work.
>
> **W5-end:** *The writing style of the paper could be improved. I report here some specific examples.*
>
> **Response:** Thank you for pointing out these edits. We will incorporate them into the final draft.
>
> **Q1:** _Would it be possible to bypass the memorization heuristic by changing both the model and the data? [...]Once a learner has trained a good model on dataset $D^*$, it should be possible to add (a large number of) redundant neurons and retrain those on D. Would this be a valid attack on PoTD?_
>
> **Response:** Your idea is a good one, and we suspect there may be a viable attack in this direction that would require new defenses. In its current form, this attack would not be resistant to segment-wise retraining, because when retraining any specific segment, either the real neurons or the fake neurons (or both) would not be correctly reproduced. An additional trick a Prover could add to this attack could be adding so many neurons that they dominate the reproduction-error term, and the “true” neurons appear as “noise” smaller than $\epsilon$. One defense would just be to require higher levels of reproducibility (e.g. small epsilon, or even perfect reproducibility if possible). We are also not sure that it is at all straightforward to get two parts of a neural network to memorize two different datasets, and never interfere with each others’ predictions; the tricks required to do so may introduce additional artifacts a Verifier could spot.
>
> **Q2:** *The Equation between lines 172 and 173 is not obvious at first sight, and should be clarified.*
>
> **Response:** We will add a step-by-step derivation in the appendix.

---

> > ### Comment · Reviewer_G7nA · 2023-08-20
> >
> > I thank the authors for the clarification.

---

### Official Review · Reviewer_yjkq · 2023-07-06

**Soundness:** 3 good
**Presentation:** 2 fair
**Contribution:** 3 good
**Rating:** 6
**Confidence:** 3

**Summary:**

The paper presents a novel protocol called Proof-of-Training-Data, which a third party auditor can verify the data used to train a model. Here, the auditor will require training data, training code, and intermediate checkpoints. Experiments on two language models have demonstrated that known attacks from the Proof-of-Learning literature can be caught by this new protocol.

**Strengths:**

This paper attempts to tackle an important security problem on trained neural network models. The proposed heuristics (i.e., memorization-based tests) is appealing and can efficiently catch spoofed checkpoints using a small amount of data. The paper is adequately structured and solid experiments have been carried out to empirically justify the effectiveness of the proposed protocol.

**Weaknesses:**

The concept of Proof-of-Learning has been well studied. Although the authors have discussed various Proof-of-Learning literature in the related work and the experiments section, it is still not immediately clear to me why we need this brand-new protocol (Proof-of-Training-Data). If I understand correctly, the authors are attempting to solve an even harder problem where the adversaries can have more computing power.

**Questions:**

- Line 77, could there be a comparison between the formal formulation of Proof-of-Learning and Proof-of-Training-Data?
- Section 4, as most of the Proof-of-Learning experiments have been done on image datasets like CIFAR, I am curious if PoTD could perform similarly well on those image datasets.


**Limitations:**

The authors have adequately addressed the limitations.

---

> ### Author Rebuttal · Authors · 2023-08-10
>
> Thank you for your feedback. Below, we respond specific points made in your review:
>
> **W1:** *It is still not immediately clear to me why we need this brand-new protocol (Proof-of-Training-Data).*
>
> **Response:** We were uncertain as to whether you were unclear about “the difference between the definitions of PoL and PoTD”, or unclear about “why existing methods from the PoL literature are insufficient to guarantee PoTD”. Just in case, we provide answers to both.
>
> First, we clarify the difference between PoL and PoTD. **PoL only checks if the Prover is capable of spending the compute** to perform a large training run, but does not check whether the training transcript disclosed **actually corresponds** to the training run that yielded $W^*$. For example, a Prover can do an original training run, and then replace 10% of the original training data with synthetic data as in [1] (e.g. to hide that data from the Verifier), and the resulting modified transcript **would still be a valid PoL.** PoTD is more ambitious in that it **checks whether the exact reported data transcript would actually result in** $W^*$. We will better highlight this distinction in the camera ready.
>
> Next, we clarify why existing methods from the PoL literature are insufficient to solve our PoTD problem:
> - **Segment-wise retraining is known to be potentially vulnerable to synthetic data attacks** [1], which allow the Prover to swap the real data out for carefully-crafted fake data. Our new verified-data-shuffling defense in Section 3.3 stops this attack.
> - **Retraining is inefficient for catching small data lies**. Given 100 checkpoints, if a Prover wants to misreport 1 checkpoint’s worth of data, then to catch them w/ 80% probability the Verifier would have to rerun 80% of the original training run. However, with our method, the Verifier needs to spend as little as 1% (and only inference compute) to catch subtraction/interpolation attacks, and slightly more for addition attacks (which require retraining a couple segments to calibrate segment-magnitudes, but still yield a roughly order-of-magnitude improvement). Even if a Verifier is only worried about Provers misreporting 10 checkpoints, the Verifier would need to retrain $\approx 16\%$ of the checkpoints (see Appendix C of [2] for a similar derivation) to catch the lying Prover with 80% probability, much larger than the 1% or few percent required by our method.
>
> We will make this direct comparison between PoL and PoTD clearer in the final version of the paper, and use the above examples to better highlight the shortcomings of prior methods at achieving PoTD.
>
> [1] Zhang, Rui, et al. "“Adversarial Examples” for Proof-of-Learning." 2022 IEEE Symposium on Security and Privacy (SP). IEEE, 2022.
>
> [2] Shavit, Yonadav. "What does it take to catch a Chinchilla? Verifying Rules on Large-Scale Neural Network Training via Compute Monitoring." arXiv preprint arXiv:2303.11341 (2023).
>
> **Q1:** *Line 77, could there be a comparison between the formal formulation of Proof-of-Learning and Proof-of-Training-Data?*
>
> **Response:** We will include this comparison, see our response to W1.
>
> **Q2:** *Section 4, as most of the Proof-of-Learning experiments have been done on image datasets like CIFAR, I am curious if PoTD could perform similarly well on those image datasets.*
>
> **Response:** As mentioned in lines 312-316, we are also interested in future work testing PoTD on new modalities. Our efforts were particularly focused on scaling PoTDs to much larger models than the traditional PoL literature, to demonstrate their practicality. Specifically, works like [3] consider ResNets on CIFAR (ResNet-50 has 25M parameters and CIFAR-10 has 60K examples and is 163 MB), whereas our results are shown LLMs with large text corpora (125M to 1B parameters and OpenWebText has ~9B tokens and is ~17GB).
>
> [3] Jia, Hengrui, et al. "Proof-of-learning: Definitions and practice." 2021 IEEE Symposium on Security and Privacy (SP). IEEE, 2021.

---

> > ### Comment · Reviewer_yjkq · 2023-08-19
> > **Thanks**
> >
> > Thanks a lot for the clarification!

---

### Official Review · Reviewer_Ejfn · 2023-07-07

**Soundness:** 2 fair
**Presentation:** 3 good
**Contribution:** 2 fair
**Rating:** 5
**Confidence:** 2

**Summary:**

This paper describes techniques and tools that can be used for verifying the "provenance" of large neural models, to evaluate their risks. These techniques and tools are part of "protocols" used by a model trainer to convince a "verifier" that the training data was used to produce the model parameters. The authors show experimentally that their prescribed procedures can catch a variety of known attacks from the "proof-of-learning" literature.

**Strengths:**

* The paper addresses an increasingly important problem, as large neural models are becoming very popular, and advances practical techniques that can be used by regulators to check the provenance of large models.

* The authors present convincing evaluation using GPT-2, demonstrating that the proposed procedures are effective in catching a variety of attacks, such as glue-ing and interpolation as well as data addition and subtraction.

**Weaknesses:**

* I find that the use of "proofs" in the title and throughout the paper is misleading as the authors do not present techniques that amount to an actual proof.

* I think it is great that the techniques presented by the authors are practical and effective wrt several attacks but I wonder if this is enough for regulators. I mean they would possibly need stronger guarantees for such techniques.

* It is unclear to me how the verification strategies presented in section 3 relate to definition 1. The authors should work on adding a theorem that clearly states that their strategies achieve the desired properties, i.e., the verifier accepts/rejects true winesses/spoofs with the desired probabilities.

* The technical contribution beyond "proof-of-learning" is unclear.

**Questions:**

Please see above.

**Limitations:**

In conclusion this is very interesting work but may be too preliminary for publication.

---

> ### Author Rebuttal · Authors · 2023-08-10
>
> Thank you for your feedback. Below, we respond specific points made in your review:
>
> **W1:** *I find that the use of "proofs" in the title and throughout the paper is misleading as the authors do not present techniques that amount to an actual proof.*
>
> **Response:** The use of the word “Proof” in Proof-of-Training-Data comes from a large body of existing literature (e.g. “Proofs-of-Learning”, which are themselves also not “proofs”).  However, we agree that this may be confusing, so we have changed the paper’s title for the final version to “Tools for Verifying Neural Models’ Training Data”.
>
> **W2:** *I think it is great that the techniques presented by the authors are practical and effective wrt several attacks but I wonder if this is enough for regulators. I mean they would possibly need stronger guarantees for such techniques.*
>
> **Response:** In developing this work, we spoke with US regulators who expressed interest in using these techniques, as they substantially improve the state of verifiability relative to the current baseline (which is blindly trusting Provers). Indeed, much real-world regulatory verification is done heuristically rather than with proofs.
>
> **W3:** *It is unclear to me how the verification strategies presented in section 3 relate to definition 1. The authors should work on adding a theorem that clearly states that their strategies achieve the desired properties, i.e., the verifier accepts/rejects true winesses/spoofs with the desired probabilities.*
>
> **Response:** Thank you for bringing this to our attention. We will add a subsection in Section 2 better clarifying Definition 1’s connection to our contribution, summarized below:
> - Definition 1 is intended to formalize the PoTD problem and defines robustness as being over all computable adversaries $\mathcal{A}$, but mathematically proving such guarantees is not yet possible given the lagging state of the NN theory literature. Indeed Jia et al.’s PoL protocol [1] itself does not make claims to provable robustness, as evidenced by attacks that have been found.
> - Instead, we follow the heuristic approach common in the PoL literature by enumerating a common-sense list of possible attacks, which provides a first approximation to the full space of attacks $\mathcal{A}$ in the definition.
> - We then show a solution that is robust to these attacks, and thereby demonstrate that our defenses heuristically satisfy Definition 1. As mentioned in lines 283-291, we hope future work will find new attacks (thereby better approximating $\mathcal{A}$) which could need to be addressed with new methods, just as the original PoL paper inspired further attacks and defenses.
>
> [1] Jia, Hengrui, et al. "Proof-of-learning: Definitions and practice." 2021 IEEE Symposium on Security and Privacy (SP). IEEE, 2021.
>
> **W4:** *The technical contribution beyond "proof-of-learning" is unclear.*
>
> **Response:** To clarify, beyond defining the new problem of Proof-of-Training-Data (which is distinct from the existing PoL problem), we also contribute the following:
> - We propose two new defenses, memorization-tests and verifiable data shuffling, which successfully address all published (and currently unaddressed) attacks on PoL. We also highlight several new attacks specific to PoTD (data addition & subtraction) and show that our methods can be effective at catching these too.
> - Our methods are substantially more efficient than prior work, which means that we are also the first to scale PoL/PoTD methods to LLM-scale using academic compute budgets. (Prior PoL papers scaled only to ResNet-50 on CIFAR.)

---

> ### Comment · Reviewer_Ejfn · 2023-08-10
>
> Thank you for your response which clarifies my questions. I will raise my score.

---

### Author Rebuttal · Authors · 2023-08-10

We thank the reviewers for their useful feedback, and are glad that many of them enjoyed the paper. We have written detailed responses to each reviewer’s comments, and thank the reviewers for their recommendations.

**For Reviewer pydU** we attach the figure referred to in our response.

---

### Decision · Program_Chairs · 2023-09-21

**Decision:**

Accept (poster)

**Comment:**

Reviewers found this paper exciting and unanimously recommended accept.